# Munc18-1 is crucial to overcome the inhibition of synaptic vesicle fusion by αSNAP

Karolina P. Stepien 🔵 [1,2,3], Eric A. Prinslow[1,2,3] & Josep Rizo 🔵 [1,2,3]

Munc18-1 and Munc13-1 orchestrate assembly of the SNARE complex formed by syntaxin-1, SNAP-25 and synaptobrevin, allowing exquisite regulation of neurotransmitter release. Non-regulated neurotransmitter release might be prevented by αSNAP, which inhibits exocytosis and SNARE-dependent liposome fusion. However, distinct mechanisms of inhibition by αSNAP were suggested, and it is unknown how such inhibition is overcome. Using liposome fusion assays, FRET and NMR spectroscopy, here we provide a comprehensive view of the mechanisms underlying the inhibitory functions of αSNAP, showing that αSNAP potently inhibits liposome fusion by: binding to syntaxin-1, hindering Munc18-1 binding; binding to syntaxin-1-SNAP-25 heterodimers, precluding SNARE complex formation; and binding to trans-SNARE complexes, preventing fusion. Importantly, inhibition by αSNAP is avoided only when Munc18-1 binds first to syntaxin-1, leading to Munc18-1-Munc13-1-dependent liposome fusion. We propose that at least some of the inhibitory activities of αSNAP ensure that neurotransmitter release occurs through the highly-regulated Munc18-1-Munc13-1 pathway at the active zone.

[1] Department of Biophysics, University of Texas Southwestern Medical Center, Dallas, TX, USA. [2] Department of Biochemistry, University of Texas Southwestern Medical Center, Dallas, TX, USA. [3] Department of Pharmacology, University of Texas Southwestern Medical Center, Dallas, TX, USA. Correspondence and requests for materials should be addressed to J.R. (email: Jose.Rizo-Rey@UTSouthwestern.edu)

Communication between neurons depends on the release of neurotransmitters by $Ca^{2+}$-triggered synaptic vesicle exocytosis. This process involves tethering of synaptic vesicles to specialized areas of the pre-synaptic plasma membrane called active zones, a priming reaction(s) that leaves the vesicles ready for release, and very fast (<1 ms) fusion of the vesicle and plasma membranes upon $Ca^{2+}$ influx into the pre-synaptic terminal[1]. These steps are exquisitely regulated during a wide variety of pre-synaptic plasticity processes that shape the properties of neural networks and underlie diverse forms of information processing in the brain[2]. Thus, although membrane fusion is a key event for neurotransmitter release, the most fundamental function of the neurotransmitter release machinery is not membrane fusion per se but to govern fusion in a precisely regulated manner that is crucial for brain function: release must occur in the right place (the active zone), at the right time (upon $Ca^{2+}$ influx), and with the right probability.

Core components of the release machinery[1,3–5] include the soluble *N*-ethylmaleimide (NEM)-sensitive factor (NSF) attachment protein (SNAP) receptors (SNAREs) syntaxin-1, SNAP-25, and synaptobrevin, which form a four-helix bundle called the SNARE complex that brings the vesicle and plasma membranes into close proximity and is key for membrane fusion[6–9]. This complex is disassembled by NSF and SNAPs to recycle the SNAREs for another round of fusion[6,10]. Munc18-1 and Munc13-1 orchestrate SNARE complex formation by an NSF-SNAP-resistant mechanism[11] whereby Munc18-1 first binds to a self-inhibited "closed" conformation of syntaxin-1[12,13] (Fig. 1, state 0) and later to synaptobrevin to template SNARE complex assembly[14–17], while Munc13-1 bridges the vesicle and plasma membranes[18,19] and helps opening syntaxin-1[20–22] (see Fig. 1). The importance of this pathway for SNARE complex assembly is illustrated by the total abrogation of neurotransmitter release observed in the absence Munc18-1[23] or Munc13s[24,25].

The tight regulation of neurotransmitter release is mediated in part by specialized factors such as the $Ca^{2+}$ sensor synaptotagmin-1 and complexins[3], but also depends on unique features of the core components that are not generally shared with their homologs. Munc13-1 has a MUN domain homologous to diverse tethering factors[26], but also contains multiple domains that are not present in other tethering factors and underlie the many regulatory functions of Munc13-1[27]. Syntaxin-1 and Munc18-1 share common domain architectures with their homologs, but the syntaxin-1 closed conformation and its complex with Munc18-1 are not general features in other forms of membrane traffic (e.g., ref. [28]). SNARE complex formation is hindered by the closed syntaxin-1 conformation and by a furled conformation of a Munc18-1 loop that prevents synaptobrevin binding[16], and the strong phenotypes observed in *Caenorhabditis elegans* lacking the invertebrate homolog of Munc13s[20,29] can be partially rescued by mutations that open syntaxin-1[12,20] or unfurl the Munc18-1 loop[14,16,29]. These results suggest that the energy barriers within the Munc18-1-syntaxin-1 complex that hinder SNARE complex assembly are crucial to render neurotransmitter release strictly dependent on Munc13s, enabling their roles as master regulators of release[3].

This beautiful design likely provides a key framework for pre-synaptic plasticity, but it is still unclear how neurotransmitter

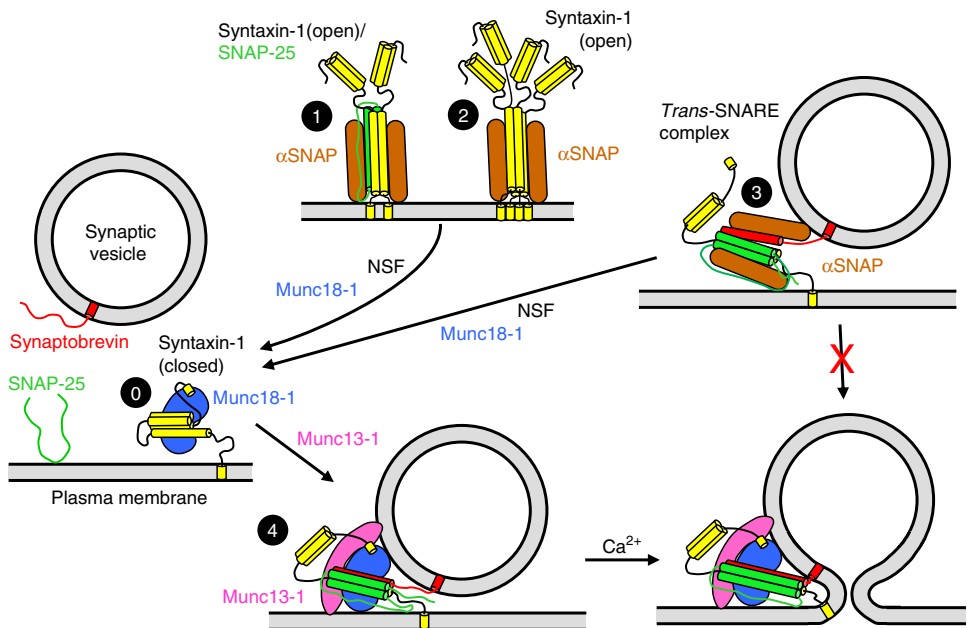

**Fig. 1** Model of how αSNAP inhibition ensures that synaptic vesicle fusion requires Munc18-1 and Munc13-1. The model postulates that syntaxin-1 can exist in different states on the pre-synaptic plasma membrane, including a state where syntaxin-1 forms a closed conformation that binds to Munc18-1 (state 0), various states where syntaxin-1 forms heterodimers with SNAP-25 (state 1, shown with a 2:1 stoichiometry) and a tetrameric state (state 2). *Trans*-SNARE complexes can be potentially formed by the SNAREs alone (state 3), or through the Munc18-1-Munc13-1-dependent pathway that starts at state 0, leading to state 4. αSNAP can inhibit synaptic vesicle fusion by binding to the syntaxin-1 tetramers, hindering Munc18-1 binding, by binding to syntaxin-1-SNAP-25 heterodimers, precluding SNARE complex assembly, and by binding to *trans*-SNARE complexes formed by SNAREs alone, preventing membrane fusion. The mechanism underlying the latter inhibition is unclear, but we speculate that binding of distinct αSNAP molecules to the SNAREs and to the apposed membranes hinders C-terminal SNARE complex zippering. In states 1–3, only two αSNAP molecules are shown for simplicity, but up to four molecules are expected to be able to bind to each SNARE four-helix bundle[53]. αSNAP cannot inhibit fusion through the pathway that starts with Munc18-1 bound to closed syntaxin-1 because this interaction impedes αSNAP binding and access of αSNAP to *trans*-SNARE complexes is obstructed by Munc18-1 and/or Munc13-1. Synaptotagmin-1 and complexin, not shown for simplicity, may also contribute to prevent αSNAP binding to *trans*-SNARE complexes. This obstruction also prevents disassembly of the *trans*-SNARE complexes by NSF-αSNAP[36] (see Discussion)

release is restricted to the Munc18-1-Munc13-dependent pathway, avoiding other less regulated pathways to synaptic vesicle fusion that could be deleterious for the proper functioning of the synapse. Thus, syntaxin-1 forms heterodimers with SNAP-25, and liposomes containing these heterodimers can fuse with liposomes containing synaptobrevin[18,30,31]. Although such fusion is abolished by NSF and αSNAP, in part because they disassemble syntaxin-1-SNAP-25 heterodimers[32], syntaxin-1 and SNAP-25 are very abundant at the plasma membrane[33] and form nanoclusters where they co-localize[34], suggesting that a fraction of syntaxin-1 molecules are bound to SNAP-25 in vivo and these heterodimers might mediate Munc18-1-Munc13-independent synaptic vesicle fusion. Indeed, syntaxin-1 and SNAP-25 on plasma membrane sheets of PC12 cells are constitutively active and can form SNARE complexes with exogenous synaptobrevin[35]. The Munc18-1-Munc13-1 pathway could be selected because Munc18-1 and Munc13-1 protect trans-SNARE complexes against disassembly by NSF-αSNAP[36] and prevent de-priming of synaptic vesicles by NEM[37], an agent that inactivates NSF. However, a fraction of trans-SNARE complexes cannot be disassembled by NSF-αSNAP in vitro[36,38] and such complexes might lead to Munc18-1-Munc13-1-independent fusion. Note also that NEM causes a small amount of evoked release in neurons lacking Munc13s, but not in those lacking Munc18-1[37]. Thus, while NSF likely contributes to favoring the Munc18-1-Munc13-depent pathway of release, it appears that there is a fundamental NSF-independent mechanism that completely prevents synaptic vesicle fusion unless Munc18-1 is present, thus underlying the total abrogation of neurotransmitter release observed in Munc18-1-knockout (KO) mice.

A potential key to understand this mechanism comes from evidence indicating that αSNAP and its yeast SNAP homolog Sec17 have functions that are independent of their role as adaptors for NSF and its yeast homolog Sec18. Sec17 inhibits yeast vacuolar fusion at an early stage and this inhibition is prevented by the HOmotypic fusion and Protein Sorting (HOPS) tethering complex, which includes the Munc18-1 homolog Vps33[39,40], although Sec17 enhances fusion at later stages[40–42]. αSNAP inhibits secretory granule and acrosomal exocytosis by a mechanism that involves inhibition of docking and syntaxin-1 binding[43–45]. αSNAP inhibited lipid mixing of syntaxin-1-SNAP-25-containing liposomes with synaptobrevin liposomes or synaptic vesicles only mildly, but strongly inhibited lipid mixing between syntaxin-1-SNAP-25 liposomes and chromaffin granules[44,46]. However, the inhibition involved interference with full SNARE complex zippering rather than with SNARE-dependent docking[46]. This conclusion was supported by single molecule data showing that αSNAP destabilizes the C terminus of the SNARE complex[47], but another single molecule study reported stabilization of C-terminal zippering by αSNAP[48], suggesting a stimulatory role. Overall, it is difficult to develop a cohesive model to explain the NSF-independent activities of αSNAP observed in these studies. Some apparently contradictory results might arise because αSNAP may act on different states of syntaxin-1, which oligomerizes in addition to forming complexes with Munc18-1 and SNAP-25[27]. Note also that much of the in vitro data was obtained with syntaxin-1 lacking its N-terminal region and, importantly, it is unknown how αSNAP function is related to those of Munc18-1 and Munc13s.

To better understand the function(s) of αSNAP in neurotransmitter release and its interplay with the other core components of the release apparatus, we use a combination of liposome fusion assays and biophysical experiments. We find that αSNAP strongly inhibits fusion by three mechanisms that involve: (i) binding to isolated syntaxin-1, which hinders binding of syntaxin-1 to Munc18-1 and SNAP-25; (ii) binding to syntaxin-1-SNAP-25 complexes, preventing formation of trans-SNARE complexes with synaptobrevin; and (iii) binding to pre-formed trans-SNARE complexes, abolishing their ability to induce fusion. These different modes of inhibition can only be bypassed when Munc18-1 binds to closed syntaxin-1, rendering fusion strictly dependent on Munc18-1 and Munc13-1. We propose that the ability of αSNAP to preclude trans-SNARE complex assembly and membrane fusion induced by assembled trans-SNARE complexes provides fundamental mechanisms to prevent synaptic vesicle fusion by constitutive pathways, ensuring that neurotransmitter release occurs through the highly regulated Munc18-1-Munc13-1 pathway. Our model is presented from the outset in Fig. 1 to facilitate understanding of our experimental design and the interpretation of our results.

## Results

**Munc18-1 overcomes the inhibition of vesicle fusion by αSNAP.** To investigate how NSF-independent αSNAP function is coupled to those of other core components of the release machinery, we used an assay that simultaneously monitors lipid and content mixing between synaptobrevin-containing liposomes (V-liposomes) and syntaxin-1-SNAP-25 liposomes (T-liposomes)[49,50] and that we have used extensively[16,18,29,51]. As described previously[18], we observed little fusion between V- and T-liposomes, but a fragment spanning the conserved C-terminal region of Munc13-1 containing its $C_1$, $C_2B$, MUN, and $C_2C$ domains (M13$C_1C_2$BMUNC$_2$C) stimulated fusion in a $Ca^{2+}$-independent manner (Fig. 2a, b, gray and black traces; Supplementary Fig. 1). Munc18-1 alone did not induce fusion but generated a $Ca^{2+}$-dependent component of fusion when included together with M13$C_1C_2$BMUNC$_2$C (Fig. 2a, b, orange and cyan traces). Addition of αSNAP abolished $Ca^{2+}$-independent fusion induced by M13$C_1C_2$BMUNC$_2$C with or without Munc18-1, but could not fully suppress the $Ca^{2+}$-dependent fusion observed when Munc18-1 and M13$C_1C_2$BMUNC$_2$C were present (Fig. 2a, b, blue and red traces). Titrations with αSNAP revealed a dose-dependent inhibition of $Ca^{2+}$-dependent fusion, while there was no $Ca^{2+}$-independent fusion at any of the αSNAP concentrations tested (Fig. 2c, d). Conversely, titrations with Munc18-1 revealed a gradual increase of fusion that maximized at 2 μM Munc18-1 (Fig. 2e, f).

These results suggest that there are two pathways to liposomes fusion in these assays. One pathway yields $Ca^{2+}$-independent fusion that is stimulated by M13$C_1C_2$BMUNC$_2$C, but not by Munc18-1, whereas the other pathway yields $Ca^{2+}$-dependent fusion that depends on both M13$C_1C_2$BMUNC$_2$C and Munc18-1. Both pathways are inhibited by αSNAP, but Munc18-1 can partially overcome the inhibition of the $Ca^{2+}$-dependent pathway. We note that the relative amount of the $Ca^{2+}$-dependent component of fusion was variable for different preparations (e.g., Supplementary Fig. 1c, d; see also Fig. 5 of ref. [18]). From additional data presented below, it became apparent that $Ca^{2+}$-independent fusion is mediated by syntaxin-1-SNAP-25 heterodimers, whereas the $Ca^{2+}$-dependent component arises from a population of T-liposomes that contain mostly syntaxin-1, and distinct amounts of this population in different preparations underlie the variability in this component. We also tested whether inclusion of synaptotagmin-1 into the synaptobrevin-containing liposomes or addition of complexin-1 altered the results of these assays, but we did not observe any noticeable effects (Supplementary Fig. 2). It is plausible that synaptotagmin-1 and/or complexin-1 might accelerate the rate of fusion, but such acceleration cannot be detected on the second time-scale characteristic of these bulk assays. Regardless of this possibility, these results show that synaptotagmin-1 and

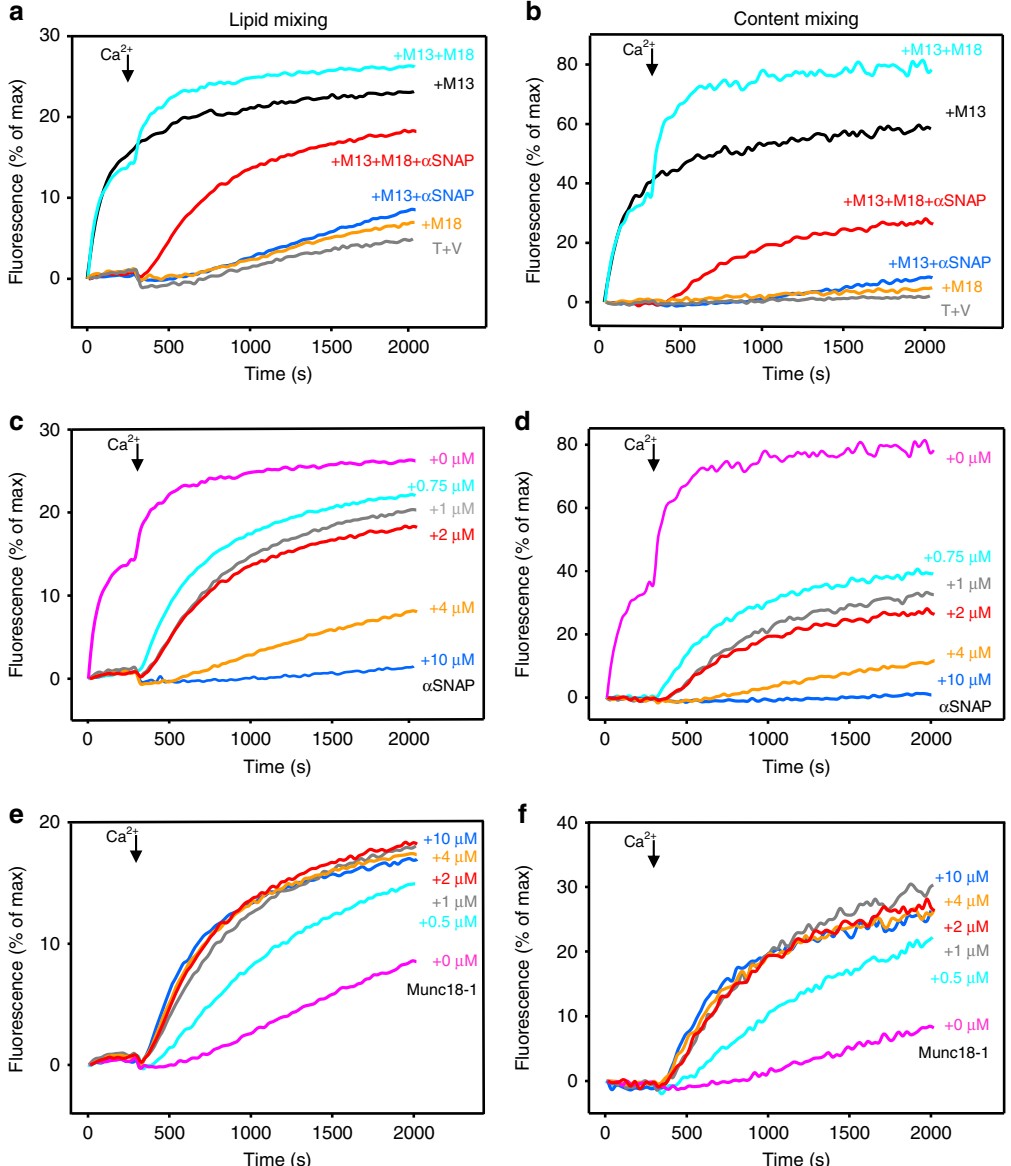

**Fig. 2** Munc18-1 partially overcomes the inhibition of fusion between T- and V-liposomes by αSNAP. **a–f** Lipid mixing (**a, c, e**) between V- and T-liposomes was monitored from the fluorescence de-quenching of Marina Blue lipids and content mixing (**b, d, f**) was monitored from the increase in the fluorescence signal of Cy5-streptavidin trapped in the V-liposomes caused by FRET with PhycoE-biotin trapped in the T-liposomes upon liposome fusion. In **a, b**, assays were performed with V- and T-liposomes alone (T + V) or including different combinations of 1 μM Munc18-1 (M18), 0.5 μM M13C$_1$C$_2$BMUNC$_2$C (M13), and 2 μM αSNAP as indicated by the color-coded labels. Assays in **c, d** included constant concentrations of M13C$_1$C$_2$BMUNC$_2$C (0.5 μM) and Munc18-1 (1 μM), and different concentrations of αSNAP, whereas assays in **e, f** included constant concentrations of M13C$_1$C$_2$BMUNC$_2$C (0.5 μM) and αSNAP (2 μM), and variable concentrations of Munc18-1. Experiments were started in the presence of 100 μM EGTA, 1 μM excess of SNAP-25, and 5 μM streptavidin, and then Ca$^{2+}$ (600 μM) was added at 300 s. Source data are provided as a Source Data file

complexin-1 cannot overcome the inhibition of fusion caused by αSNAP.

Since the role of αSNAP in SNARE complex disassembly requires binding to the SNARE four-helix bundle and is enhanced by binding to membranes[52], we examined the importance of these properties for the inhibitory activity of αSNAP. The ability of αSNAP to inhibit Ca$^{2+}$-independent and Ca$^{2+}$-dependent fusion was strongly impaired by a double charge reversal that hinders SNARE binding[53] (K122E,K163E; KE mutant) or a double replacement of hydrophobic residues that disrupts membrane binding[52] (F27S,F28S; FS mutant) (Fig. 3, Supplementary Fig. 3), showing that SNARE binding and membrane binding are indeed key for the inhibitory activities of αSNAP.

In the presence of ATP-bound NSF together with αSNAP, fusion required Munc18-1 and M13C$_1$C$_2$BMUNC$_2$C, and was Ca$^{2+}$-dependent (Fig. 4a, b cyan traces), as previously described[18]. Such fusion was much more efficient than that observed in the presence of the non-hydrolyzable ATP analog ATPγS (Fig. 4a, b, orange traces), which yielded similar results as those obtained with Munc18-1, M13C$_1$C$_2$BMUNC$_2$C, and αSNAP in the absence of NSF (Fig. 2a, b, red traces). These results suggest that the inhibition of fusion caused by αSNAP is released by NSF and that such release requires ATP hydrolysis by NSF, but under these conditions fusion requires Ca$^{2+}$. To test whether NSF can release the inhibition after fusion has been arrested by αSNAP, we performed fusion assays where reagents

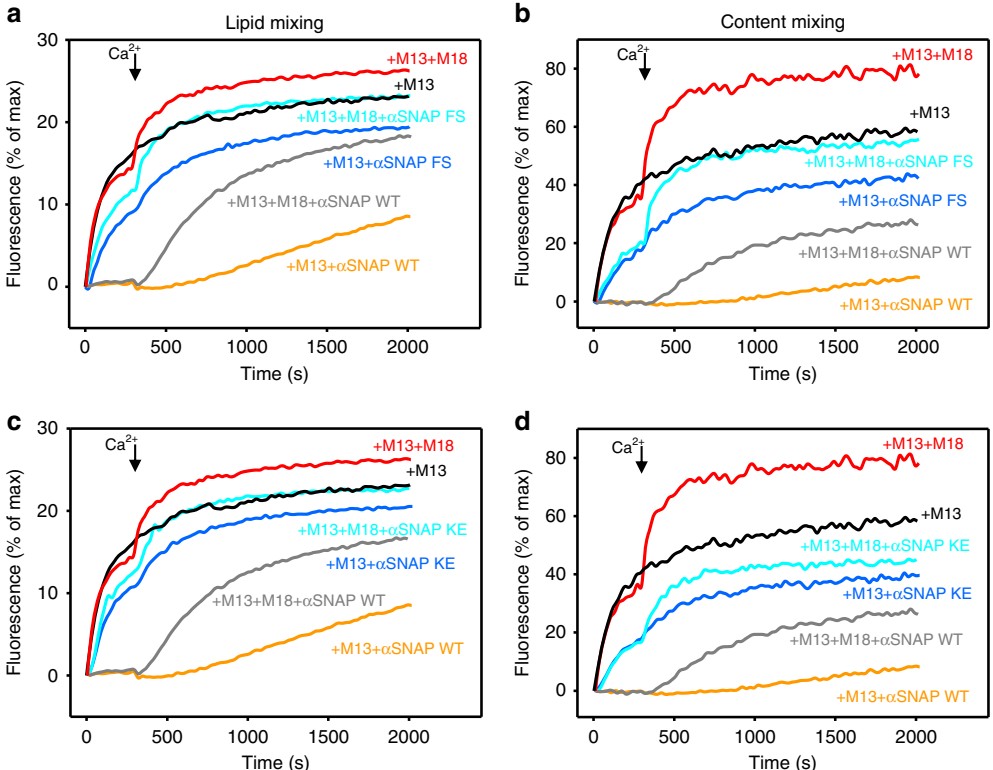

**Fig. 3** Binding of αSNAP to the SNAREs and membranes is critical for its inhibitory activity. **a–d** Lipid mixing (**a**, **c**) between V- and T-liposomes was monitored from the fluorescence de-quenching of Marina Blue lipids and content mixing (**b**, **d**) was monitored from the increase in the fluorescence signal of Cy5-streptavidin trapped in the V-liposomes caused by FRET with PhycoE-biotin trapped in the T-liposomes upon liposome fusion. Assays were performed in the presence of 0.5 μM M13$C_1C_2$BMUNC$_2$C (M13) with or without 1 μM Munc18-1 (M18) and with 2 μM WT or mutant αSNAP (αSNAP FS mutant in **a**, **b**; αSNAP KE mutant in **c**, **d**). Experiments were started in the presence of 100 μM EGTA, 1 μM excess of SNAP-25, and 5 μM streptavidin, and the Ca$^{2+}$ (600 μM) was added at 300 s. Source data are provided as a Source Data file

were added sequentially, and NSF was added last. Mixing T- and V-liposomes with Munc18-1 yielded very little fusion, but fusion was strongly stimulated by addition of M13$C_1C_2$BMUNC$_2$C and the progress of fusion was immediately stopped by αSNAP (Fig. 4c, d, black traces). Subsequent addition of Ca$^{2+}$ led to a slight increase in fusion, but final addition of NSF led to a fast increase in fusion (Fig. 4c, d, black traces), reaching similar maximum levels as those observed in standard fusion assays that included all these proteins from the beginning (e.g., Figure 4a, b, cyan traces). In parallel experiments where we added the same reagents sequentially but excluding Munc18-1, we observed a comparable increase in fusion when M13$C_1C_2$BMUNC$_2$C was added and fusion was also stopped by αSNAP, but there was no increase in fusion upon addition of Ca$^{2+}$ and NSF (Fig. 4c, d, green traces). These results show that NSF releases the inhibition of fusion caused by αSNAP, but release of this inhibition requires Munc18-1.

**Vesicle fusion pathways depend on the state of syntaxin-1**. The closed conformation of syntaxin-1 is formed by intramolecular binding of the N-terminal H$_{abc}$ domain to the SNARE motif involved in forming the SNARE complex (see domain diagram in Fig. 5a, and models of Fig. 1, states 0 and 4)[12,13]. The closed conformation hinders other interactions involving the syntaxin-1 SNARE motif, including those underlying not only SNARE complex assembly but also oligomerization, binding to αSNAP[53,54] and formation of heterodimers with SNAP-25 that most often have a 2:1 (syntaxin-1:SNAP-25) stoichiometry[27] (Fig. 1, states 1 and 2). To gain insight into the role of the closed conformation and to understand the mechanisms involved in

αSNAP inhibition (Fig. 2a, b), we performed fusion assays using T-liposomes that contained a syntaxin-1 fragment lacking the H$_{abc}$ domain (ΔH$_{abc}$) (Fig. 5a). In fusion assays including Munc18-1, M13$C_1C_2$BMUNC$_2$C, NSF, and αSNAP, deletion of H$_{abc}$ abolished fusion (Fig. 5b, c, brown traces; Supplementary Fig. 4a, b), consistent with the strong impairment of synaptic vesicle priming caused by this deletion[55]. The amounts of syntaxin-1 ΔH$_{abc}$ and SNAP-25 incorporated into the T-liposomes were comparable to those of T-liposomes containing wild-type (WT) syntaxin-1 (Supplementary Fig. 4c), showing that this effect did not arise from poor protein incorporation during reconstitution. In fusion assays performed without NSF, syntaxin-1 ΔH$_{abc}$ supported Ca$^{2+}$-independent fusion in the presence of M13$C_1C_2$BMUNC$_2$C (Fig. 5d, e, black traces), similar to experiments conducted with WT syntaxin-1 (Fig. 2a, b, black traces, Supplementary Fig. 4d, e). However, there was little Ca$^{2+}$-dependent increase in fusion in the presence of M13$C_1C_2$BMUNC$_2$C and Munc18-1 (Fig. 5d, e, cyan traces). The addition of αSNAP completely abolished fusion in the presence of M13$C_1C_2$BMUNC$_2$C with or without Munc18-1 (Fig. 5d, e, blue and red traces), showing that Munc18-1 cannot overcome the inhibition caused by αSNAP when syntaxin-1 lacks the H$_{abc}$ domain.

To interpret our results, it is important to consider that NSF and αSNAP disassemble different types of four-helix bundles formed by the SNAREs, including syntaxin-1-SNAP-25 heterodimers[32], *cis*-SNARE complexes in solution and on membranes[6,52], and *trans*-SNARE complexes between two membranes[36,38]. Moreover, αSNAP was found to bind tightly to syntaxin-1, but this complex was disassembled by NSF and could not re-assemble afterwards[54], suggesting that initial binding of

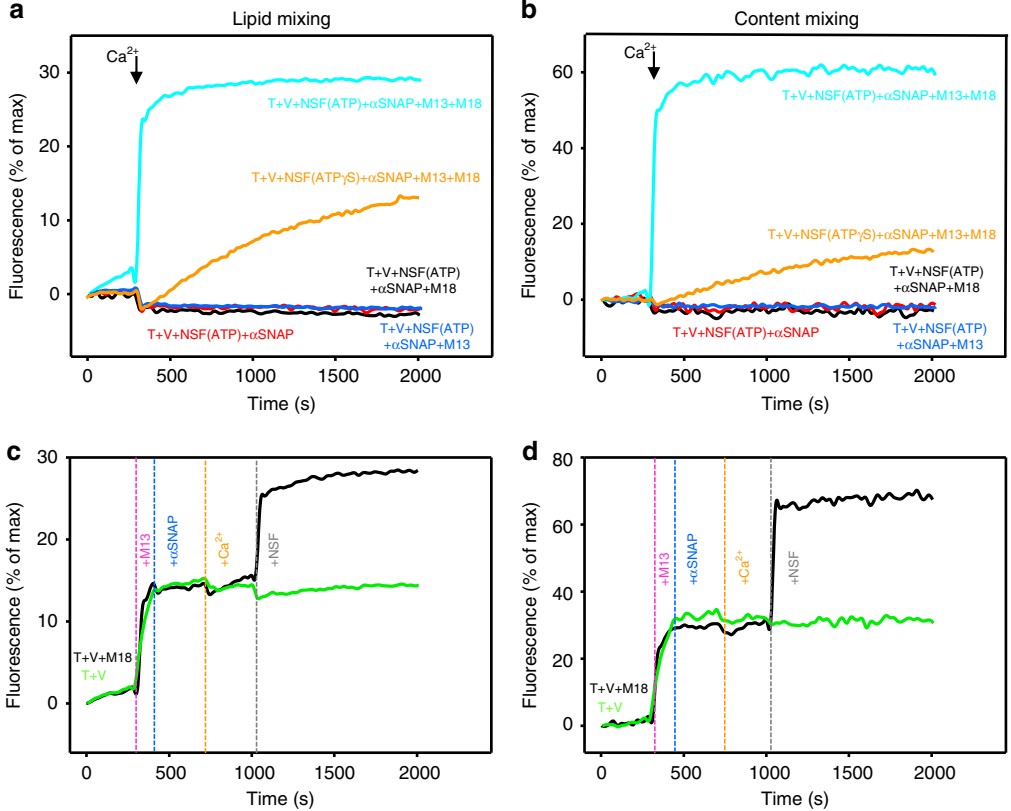

**Fig. 4** NSF rescues the inhibition of fusion between V- and T-liposomes caused by αSNAP. **a, b** Lipid mixing (**a**) between V- and T-liposomes was monitored from the fluorescence de-quenching of Marina Blue lipids and content mixing (**b**) was monitored from the increase in the fluorescence signal of Cy5-streptavidin trapped in the V-liposomes caused by FRET with PhycoE-biotin trapped in the T-liposomes upon liposome fusion. Assays were performed in the presence of 0.8 μM NSF and 2 μM αSNAP without or with 1 μM Munc18-1 (M18) and/or 0.5 μM M13C₁C₂BMUNC₂C (M13) as indicated by the color-coded labels. All traces were acquired with NSF in the presence of ATP, except the orange traces, where ATP was replaced by ATPγS. Experiments were started in the presence of 100 μM EGTA, 1 μM excess of SNAP-25, and 5 μM streptavidin, and Ca²⁺ (600 μM) was then added at 300 s. **c, d** Analogous assays where reagents were added sequentially. Experiments were started in the presence (black trace) or absence (green trace) of 1 μM Munc18-1; 0.5 μM M13C₁C₂BMUNC₂C, 2 μM αSNAP, 600 μM Ca²⁺, and 0.8 μM NSF were added at the indicated time points. Note that, in the presence of Munc18-1, M13C₁C₂BMUNC₂C strongly stimulated fusion, αSNAP immediately stopped fusion, Ca²⁺ induced a small amount of fusion that varied in different experiments, and NSF sharply restored fusion. No such recovery was observed in the absence of Munc18-1. Source data are provided as a Source Data file

αSNAP involved a syntaxin-1 oligomer, likely a tetramer, whereas syntaxin-1 remained monomeric and closed (thus unable to bind to αSNAP) after disassembly by NSF. As four molecules of αSNAP can bind around SNARE four-helix bundles, covering most of their surface[53], such binding should hinder interactions of the four-helix bundles with other proteins. Based on these observations, our results can be explained by a model that has three premises. First, Ca²⁺-independent fusion between T- and V-liposomes induced by M13C₁C₂BMUNC₂C (Fig. 2a, b, black traces) is mediated by syntaxin-1-SNAP-25 heterodimers residing on the T-liposomes, and αSNAP inhibits such fusion because it binds to the heterodimers (Fig. 1, state 1), hindering their interaction with synaptobrevin to form SNARE complexes. M13C₁C₂BMUNC₂C likely stimulates this Ca²⁺-independent pathway by bridging V- and T-liposomes[18,19] and catalyzing SNARE complex assembly via interactions with the SNARE motifs[56]. This pathway is not strongly influenced by deletion of the H_abc domain (Fig. 5d, e, black traces), which should not prevent binding of syntaxin-1 to SNAP-25.

The second premise of our model is that the Ca²⁺-dependent component of liposome fusion (Fig. 2a, b, cyan traces) is mediated by syntaxin-1 molecules that are able to bind to Munc18-1 because they can dissociate from SNAP-25 or because they are not bound to SNAP-25, perhaps due to inefficient incorporation of SNAP-25 in a subset of T-liposomes. αSNAP inhibits this component of fusion because it binds to syntaxin-1 (Fig. 1, state 2), competing with Munc18-1. Because of such competition, the level of fusion depends on the relative amounts of Munc18-1 and αSNAP (Fig. 2c–f). Fusion is Ca²⁺-dependent in this case because there is a high energy barrier to SNARE complex assembly when starting with the syntaxin-1-Munc18-1 complex, and fusion requires Ca²⁺ binding to the Munc13-1 C₂B domain, which makes M13C₁C₂BMUNC₂C more active[16,18,36,51]. For syntaxin-1 ΔH_abc, the inhibition of fusion caused by αSNAP cannot be overcome by Munc18-1 (Fig. 5d, e, red traces) because the H_abc domain is essential for the closed conformation and for tight binding to Munc18-1. Finally, the third premise of our model is that NSF promotes Ca²⁺-dependent fusion in the presence of Munc18-1, M13C₁C₂BMUNC₂C, and αSNAP (Fig. 4a, b, cyan traces) by disassembling the complexes that αSNAP forms with SNARE four-helix bundles, leading to the formation of closed syntaxin-1 monomers and thus guiding the system to the Munc18-1-Munc13-1 pathway of fusion (Fig. 1).

**Munc18-1 binding to closed syntaxin-1 is crucial for fusion.** Our first premise predicts that αSNAP inhibits fusion starting with syntaxin-1-SNAP-25 heterodimers on T-liposomes by binding to these heterodimers, thus hindering formation of *trans*-SNARE complexes with synaptobrevin on V-liposomes.

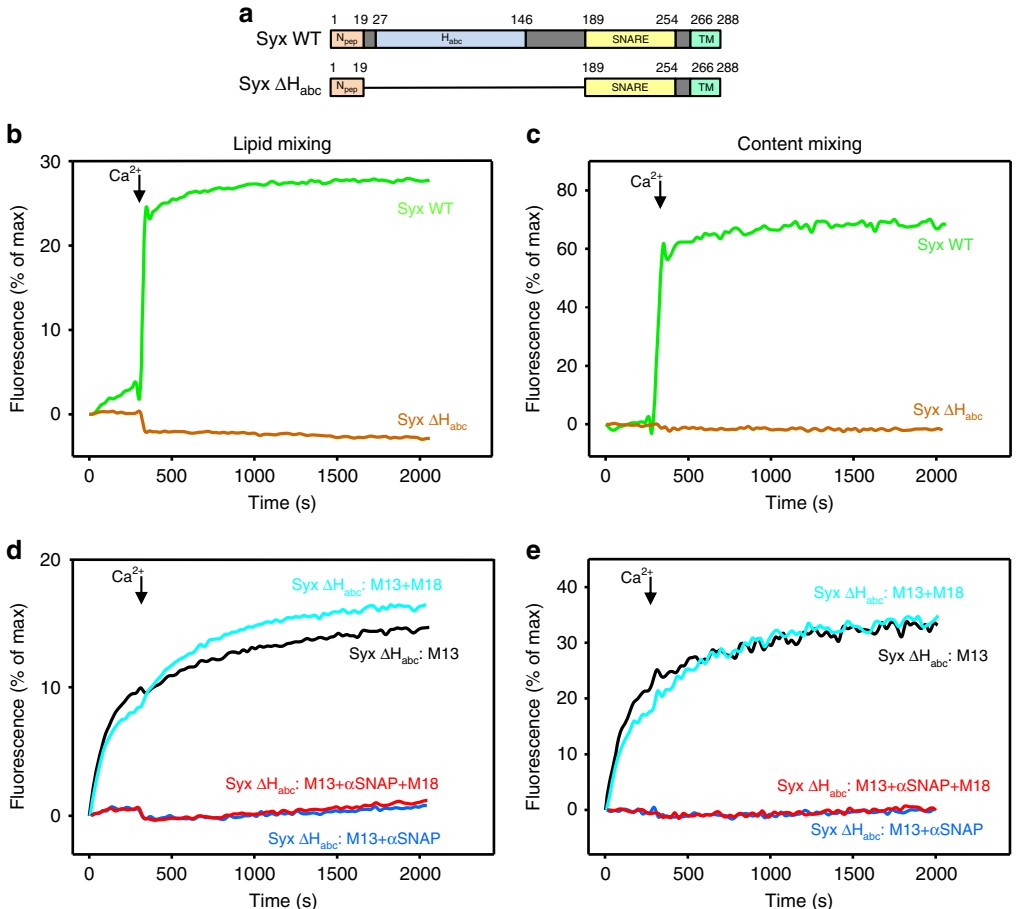

**Fig. 5** Binding of Munc18-1 to closed syntaxin-1 is key to overcome the inhibition of fusion by αSNAP. **a** Domain diagrams of syntaxin-1 showing the design of the syntaxin-1 mutant lacking the N-terminal $H_{abc}$ domain ($\Delta H_{abc}$). $N_{pep}$ denotes the N-peptide, $H_{abc}$ the $H_{abc}$ domain, SNARE the SNARE motif, and TM the transmembrane region. **b**–**e** Lipid mixing (**b**, **d**) between V- and T-liposomes was monitored from the fluorescence de-quenching of Marina Blue lipids and content mixing (**c**, **e**) was monitored from the increase in the fluorescence signal of Cy5-streptavidin trapped in the V-liposomes caused by FRET with PhycoE-biotin trapped in the T-liposomes upon liposome fusion. In **b**, **c**, assays were performed in the presence of 0.8 μM NSF, 2 μM αSNAP, 1 μM Munc18-1 (M18), and 0.5 μM M13$C_1C_2$BMUNC$_2$C (M13) using T-liposomes that contained WT syntaxin-1 (Syx WT) or the syntaxin-1 mutant lacking the Habc domain ($\Delta H_{abc}$). In **d**, **e**, experiments were performed with T-liposomes containing $\Delta H_{abc}$ syntaxin-1 mutants in the presence of 0.5 μM M13$C_1C_2$BMUNC$_2$C with or without 1 μM Munc18-1 and with or without 2 μM αSNAP. Experiments were started in the presence of 100 μM EGTA, 1 μM excess of SNAP-25, and 5 μM streptavidin, and then $Ca^{2+}$ (600 μM) was added at 300 s. Source data are provided as a Source Data file

This notion is supported by data showing that αSNAP inhibits formation of *cis*-SNARE complexes containing a syntaxin-1 fragment lacking its N-terminal region[46]. To test this premise in a more realistic setting, we used a fluorescence resonance energy transfer (FRET) assay that we recently developed to monitor assembly of *trans*-SNARE complexes between T- and V-liposomes using full-length SNAREs, with a mutation in SNAP-25 to prevent fusion[36]. Since *trans*-SNARE complex assembly is very slow because of the formation 2:1 syntaxin-1-SNAP-25 heterodimers (reviewed in ref. [27]), we accelerated *trans*-SNARE complex assembly by including Munc13-1 $C_1C_2$BMUNC$_2$C, which bridges V- and T-liposomes[19], and a C-terminal peptide spanning the C-terminal half of synaptobrevin (Syb49–93), which displaces the second syntaxin-1 molecule from the heterodimers, facilitating binding to synaptobrevin[57] (Fig. 6a). As expected, we observed efficient *trans*-SNARE complex assembly in the presence of $C_1C_2$BMUNC$_2$C and Syb49–93, but inclusion of αSNAP potently inhibited assembly (Fig. 6b, black and blue traces). When αSNAP was added in the middle of the reaction, the progress of *trans*-SNARE complex assembly was immediately stopped (Fig. 6b, red trace), showing that the inhibition

caused by αSNAP occurs much faster than the assembly reaction.

The second premise of our model implies that $Ca^{2+}$-dependent fusion between T- and V-liposomes induced by Munc18-1 and M13$C_1C_2$BMUNC$_2$C (Fig. 2a, b, cyan traces) starts with the closed syntaxin-1-Munc18-1 complex and that the amount of inhibition caused by αSNAP depends on the relative affinity of Munc18-1 and αSNAP for syntaxin-1, as well as the relative on-rates of the two interactions. To investigate the interplay between Munc18-1 and αSNAP without the interference of pre-assembled syntaxin-1-SNAP-25 heterodimers, we prepared liposomes containing only syntaxin-1 (S-liposomes). Interestingly, we did not observe any fusion between S- and V-liposomes in the presence of SNAP-25 and M13$C_1C_2$BMUNC$_2$C (Fig. 7a, b, black traces), which contrasts with the efficient fusion observed between V- and T-liposomes in the presence of M13$C_1C_2$BMUNC$_2$C (Fig. 2a, b, black traces). This observation suggests that the syntaxin-1 molecules present on the S-liposomes are unable to bind to SNAP-25. To test this proposal, we used $^1$H-$^{15}$N heteronuclear single quantum coherence (HSQC) spectra, which are NMR experiments that are very sensitive to protein–protein interactions[58]. Addition of S-liposomes did not substantially alter the

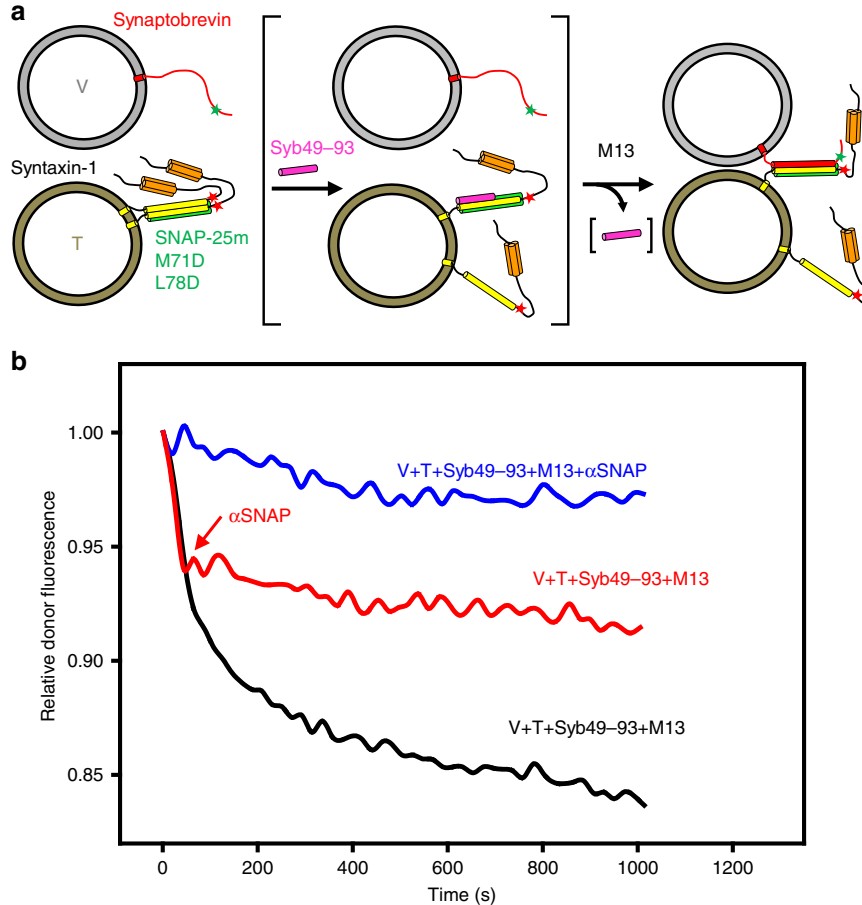

**Fig. 6** αSNAP inhibits *trans*-SNARE complex assembly. **a** Diagram illustrating the FRET assay used to monitor trans-SNARE complex assembly[36]. V-liposomes containing synaptobrevin labeled with a FRET donor (Alexa-488, green star) at residue 26 are mixed with T-liposomes containing syntaxin-1 labeled at residue 186 with a FRET acceptor (tetramethylrhodamine, red star) and SNAP-25 bearing a M71D,L78D mutation that prevents membrane fusion. Synaptobrevin is red, SNAP-25 M71D,L78D green, and syntaxin-1 orange (N-terminal $H_{abc}$ domain) and yellow (SNARE motif and TM region). Syntaxin-1-SNAP-25 M71D,L78D complexes are expected to have a 2:1 stoichiometry (left diagram), hindering SNARE complex formation, and the addition of a synaptobrevin fragment spanning the C-terminal part of its SNARE motif (residues 49–93; Syb49–93) accelerates SNARE complex assembly because it displaces the second syntaxin-1 molecule from the syntaxin-1-SNAP-2 M71D,L78D heterodimer, yielding the intermediate shown between brackets[57]. Syb49–93 can be displaced by synaptobrevin on the V-liposomes to form *trans*-SNARE complexes (right diagram). **b** Kinetic assays monitoring *trans*-SNARE complex assembly upon mixing V- and T-liposomes (1:4 ratio) in the presence of 10 μM Syb49–93 and 0.3 μM $M13C_1C_2BMUNC_2C$ (M13) with (blue trace) or without (black trace) 2 μM αSNAP. The red trace shows an analogous experiment where 2 μM αSNAP was not included initially but was added at ~200 s, which immediately stopped *trans*-SNARE complex assembly. In all assays, the fluorescence emission intensity of the donor was monitored as a function of time and the data were normalized with the intensity observed in the first point. Source data are provided as a Source Data file

$^1$H-$^{15}$N HSQC spectrum of $^{15}$N-labeled SNAP-25, whereas a soluble fragment spanning the cytoplasmic region of syntaxin-1 (Syx2–253) caused dramatic changes, as expected for binding between these two proteins (Supplementary Fig. 5). These results conclusively show that SNAP-25 cannot bind to syntaxin-1 reconstituted in these liposomes, which correlates with previous results obtained with syntaxin-1 incorporated into detergent micelles[59]. Note that in previous studies we did observe binding of SNAP-25 to syntaxin-1 liposomes[11]. These distinct results may arise because we are now purifying syntaxin-1 with a method that prevents formation of large aggregates and was used for the experiments in detergent micelles[59].

We observed efficient $Ca^{2+}$-dependent fusion between S- and V-liposomes when SNAP-25 was added together with both Munc18-1 and $M13C_1C_2BMUNC_2C$, and αSNAP inhibited fusion strongly but not completely (Fig. 7a, b, orange and pink traces; Supplementary Fig. 6). To examine whether the results depend on differential kinetics of binding of Munc18-1 and αSNAP to syntaxin-1, we performed fusion assays that included

SNAP-25, Munc18-1, $M13C_1C_2BMUNC_2C$, and αSNAP in all cases, but where the S-liposomes were pre-incubated with a different subset of these reagents. Pre-incubation with Munc18-1 led to robust $Ca^{2+}$-dependent fusion (Fig. 7c, d, gray traces), while pre-incubation of the S-liposomes with αSNAP led to almost no fusion, and no fusion was observed when the S-liposomes were pre-incubated with SNAP-25 and αSNAP (Fig. 7c, d, blue and pink traces). Pre-incubation with Munc18-1 and αSNAP yielded intermediate inhibition (Fig. 7c, d, orange traces). These results show that Munc18-1 and $M13C_1C_2BMUNC_2C$ can mediate highly efficient $Ca^{2+}$-dependent fusion between S- and V-liposomes in the presence of SNAP-25 without the assistance of the disassembly activity of NSF-αSNAP and that such fusion is inhibited by αSNAP to different extents that depend on whether syntaxin-1 binds first to Munc18-1 or to αSNAP. Hence, these data strongly support the second premise of our model and the notion that binding of Munc18-1 to closed syntaxin-1 is critical to prevent inhibition of fusion by αSNAP. Importantly, efficient $Ca^{2+}$-dependent fusion was observed in analogous

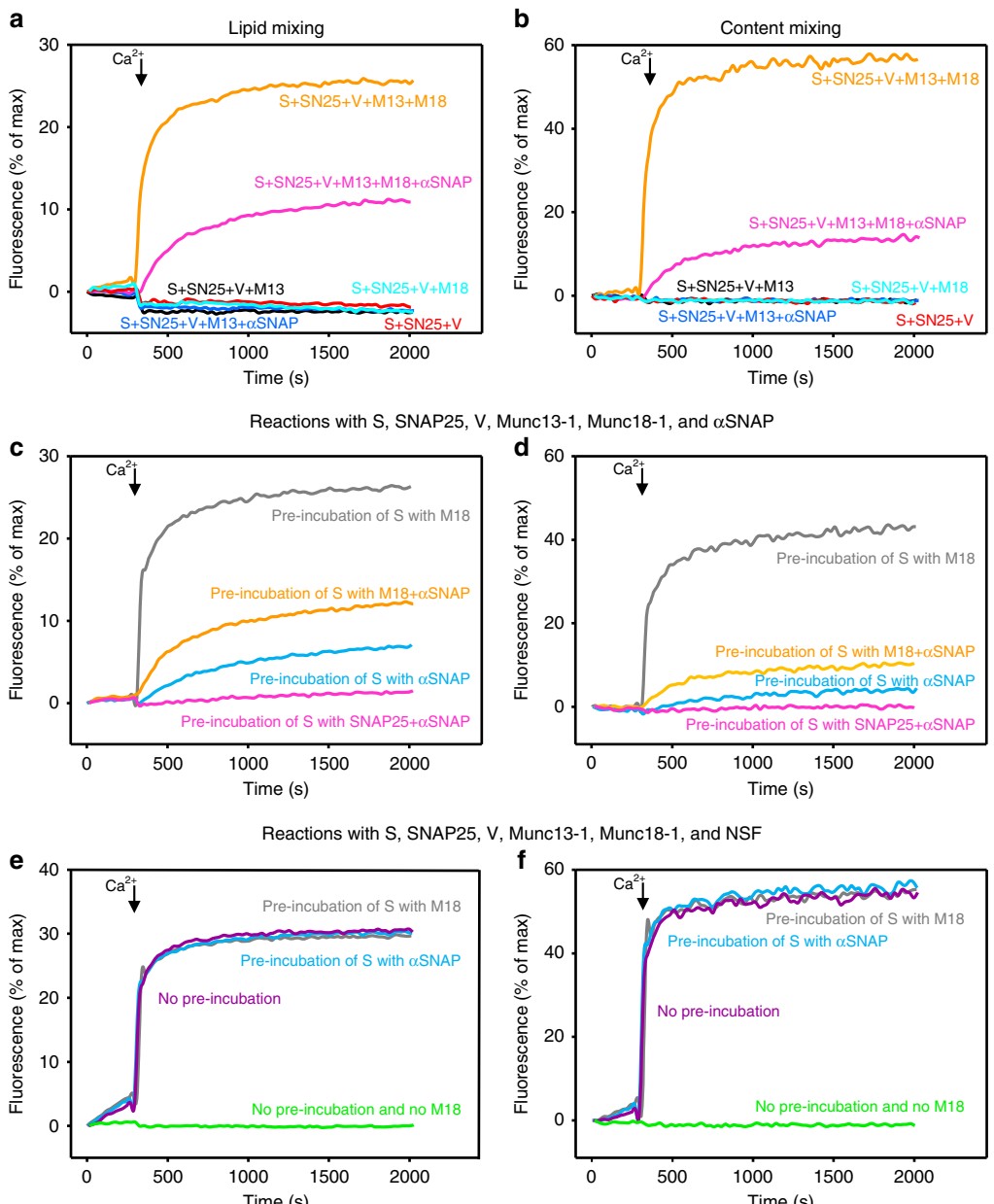

**Fig. 7** Binding of Munc18-1 to syntaxin-1 overcomes the inhibition of fusion caused by αSNAP. **a–f** Lipid mixing (**a**, **c**, **e**) between V- and S-liposomes was monitored from the fluorescence de-quenching of Marina Blue lipids and content mixing (**b**, **d**, **f**) was monitored from the increase in the fluorescence signal of Cy5-streptavidin trapped in the V-liposomes caused by FRET with PhycoE-biotin trapped in the T-liposomes upon liposome fusion. In **a**, **b**, all assays were performed in the presence of 1 μM SNAP-25 plus different combinations of 0.5 μM M13C$_1$C$_2$BMUNC$_2$C (M13), 1 μM Munc18-1 (M18), and 2 μM αSNAP. In **c**, **d**, all assays were performed in the presence of 1 μM SNAP-25, 0.5 μM M13C$_1$C$_2$BMUNC$_2$C (M13), 1 μM Munc18-1 (M18), and 2 μM αSNAP, but pre-incubating the S-liposomes with different reagents as indicated. In **e**, **f**, all assays were performed in the presence of 0.8 μM NSF, 1 μM SNAP-25, 0.5 μM M13C$_1$C$_2$BMUNC$_2$C (M13), 1 μM Munc18-1 (M18), and 2 μM αSNAP, but pre-incubating the S-liposomes with different reagents as indicated. Source data are provided as a Source Data file

experiments that included NSF in addition to Munc18-1, M13C$_1$C$_2$BMUNC$_2$C, and αSNAP, regardless of which reagent was pre-incubated with the S-liposomes (Fig. 7e, f). These results show that NSF resets the machinery, guiding the system to the Munc18-1-Munc13-1 pathway of membrane fusion by rapidly disassembling complexes arrested by αSNAP binding, as predicted by the third premise of our model.

**Munc18-1 and αSNAP bind to distinct syntaxin-1 conformations**. To investigate whether Munc18-1 and αSNAP bind to

distinct conformations of membrane-anchored syntaxin-1, we again used nuclear magnetic resonance (NMR) spectroscopy. We prepared soluble syntaxin-1(2–253) fragment specifically $^1$H,$^{13}$C-labeled at isoleucine δ1-methyl groups ($^2$H-I-$^{13}$CH$_3$-labeled) and S-liposomes containing $^2$H-I-$^{13}$CH$_3$-labeled syntaxin-1, and acquired $^1$H-$^{13}$C heteronuclear multiple quantum coherence (HMQC) spectra, which exhibit high sensitivity even for large protein complexes[60]. Although the large size of liposomes (100 MDa range) is expected to broaden beyond detection the resonances of proteins anchored on them, those of membrane-anchored syntaxin-1 may be observable if there is sufficient

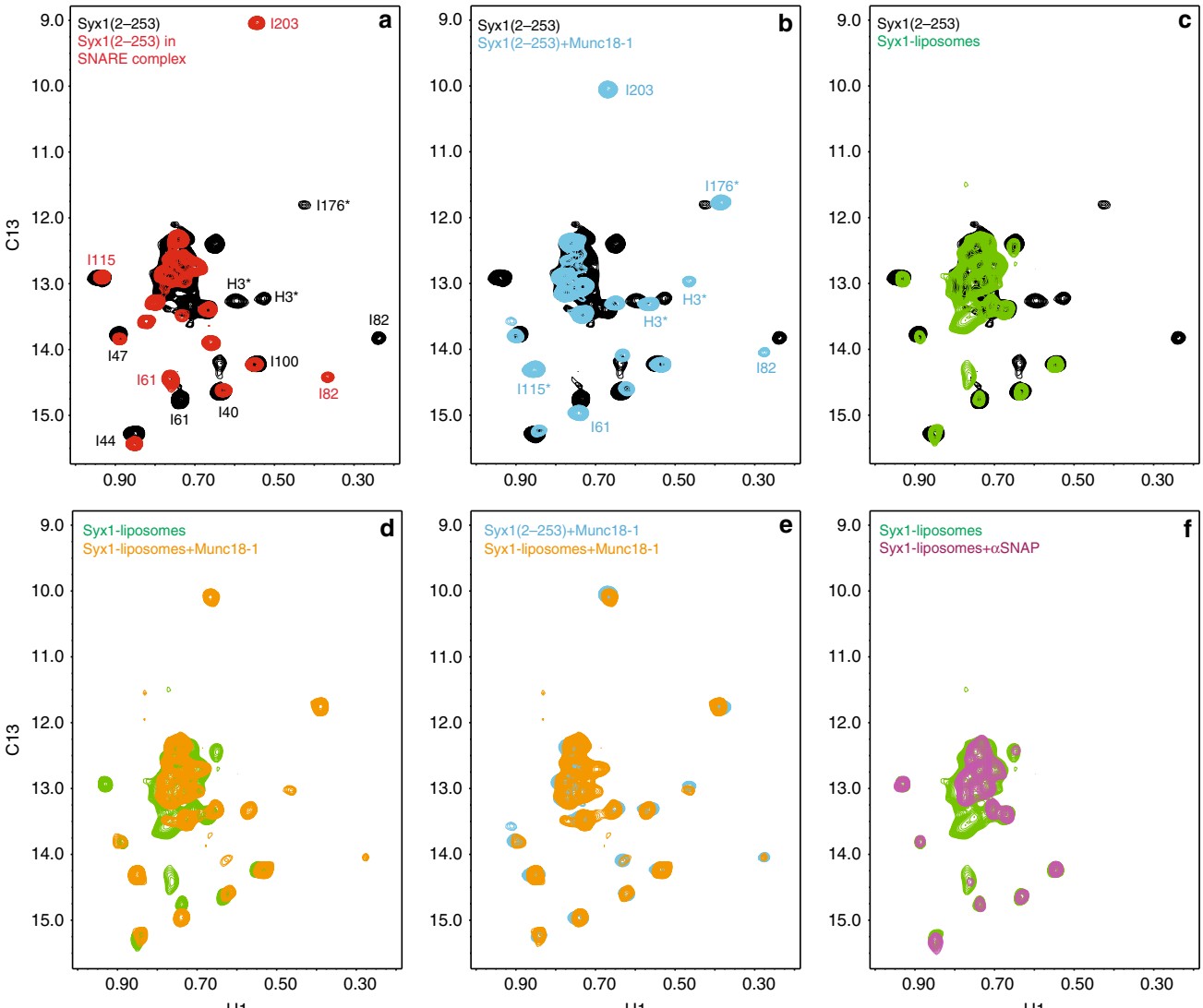

**Fig. 8** Munc18-1 and αSNAP bind to different conformations of liposome-anchored syntaxin-1. The contour plots show superpositions of $^1$H-$^{13}$C HMQC spectra of: **a** $^2$H-I-$^{13}$CH$_3$-syntaxin-1(2–253) alone (black contours) or incorporated into the SNARE complex with the SNARE motifs of synaptobrevin and SNAP-25 (red contours); **b** $^2$H-I-$^{13}$CH$_3$-syntaxin-1(2–253) alone (black contours) or bound to Munc18-1 (blue contours); **c** $^2$H-I-$^{13}$CH$_3$-syntaxin-1(2–253) (black contours) and liposomes containing $^2$H-I-$^{13}$CH$_3$-syntaxin-1 (green contours); **d** liposomes containing $^2$H-I-$^{13}$CH$_3$-syntaxin-1 alone (green contours) or bound to Munc18-1 (orange contours); **e** $^2$H-I-$^{13}$CH$_3$-syntaxin-1(2–253) bound to Munc18-1 (blue contours) and liposomes containing $^2$H-I-$^{13}$CH$_3$-syntaxin-1 bound to Munc18-1 (orange contours); and **f** liposomes containing $^2$H-I-$^{13}$CH$_3$-syntaxin-1 alone (green contours) or bound to αSNAP (purple contours). In **a**, **b**, cross-peak assignments that are available[21,62,63] are indicated. Most of these assignments correspond to the H$_{abc}$ domain. In addition, three cross-peaks were tentatively assigned to I176, which is located in the linker between the H$_{abc}$ domain and the SNARE motif, and to isoleucines in the SNARE motif (labeled H3)[21]. The * indicates the tentative nature of these assignments

flexibility in the SNARE motif and/or the linker between the H$_{abc}$ domain and the SNARE motif, much as we observed resonances of liposome-anchored synaptobrevin[61]. Assignments of some syntaxin-1 isoleucine cross-peaks are available[21,62,63]. Comparison of $^1$H-$^{13}$C HMQC spectra of syntaxin-1(2–253) alone or incorporated into the SNARE complex showed how a few cross-peaks from the H$_{abc}$ domain shift because of syntaxin-1 opening upon SNARE complex assembly (Fig. 8a). A distinctive feature of the syntaxin-1(2–253) spectrum is a well-resolved cross-peak tentatively assigned to I176 of the linker region[21], which is not well resolved for the SNARE complex because the linker is unstructured, but is observed for syntaxin-1 bound to Munc18-1 (Fig. 8b). Two additional cross-peaks tentatively assigned to isoleucines of the SNARE motif (labeled H3) are also characteristic of closed syntaxin-1 and are observable for syntaxin-1 bound to

Munc18-1, but not for the SNARE complex. These cross-peaks provide diagnostics to assess whether syntaxin-1 is open or closed. Note that the shifts induced by Munc18-1 in these three cross-peaks and some of the cross-peaks from the H$_{abc}$ domain are a natural consequence of local structural changes caused by this tight interaction.

Importantly, we were able to observe multiple cross-peaks in the $^1$H-$^{13}$C HMQC spectrum of S-liposomes containing $^2$H-I-$^{13}$CH$_3$-syntaxin-1, but none of the diagnostic cross-peaks for the closed conformation were observable, suggesting that syntaxin-1 is open on membranes (Fig. 8c). Binding of Munc18-1 caused marked changes in the spectrum of liposome-anchored syntaxin-1 (Fig. 8d), leading to a spectrum that largely coincides with that of syntaxin-1(2–253) bound to Munc18-1 (Fig. 8e). Hence, liposome-anchored syntaxin-1 adopts the closed conformation

upon binding to Munc18-1. In contrast, αSNAP induced less prominent changes in the $^1$H-$^{13}$C HMQC spectra of liposome-anchored syntaxin-1 and the cross-peaks of the H$_{abc}$ domain were largely unaffected (Fig. 8f), showing that αSNAP binds to an open conformation of syntaxin-1. Note that the default conformation of the cytoplasmic region of syntaxin-1 as a monomer is the closed conformation, and the conformation opens when the SNARE motif forms a complex with other SNAREs or oligomerizes, most likely forming a four-helix bundle[12,27,63]. Thus, our NMR data suggest that liposome-anchored syntaxin-1 is tetrameric, which explains the finding that it does not bind to SNAP-25, but readily binds to αSNAP. Munc18-1 can still bind with high affinity to syntaxin-1, but binding requires transition to the closed conformation and hence is slow compared to αSNAP binding. This model is consistent with the finding that the inhibition by αSNAP dominated in fusion assays between S- and V-liposomes where αSNAP and Munc18-1 were added from the beginning, but not when Munc18-1 was pre-incubated with the S-liposomes (Fig. 7a–d).

We attempted to further test this model by acquiring $^1$H-$^{13}$C HMQC spectra of S-liposomes that were pre-incubated with αSNAP and then mixed with Munc18-1, or vice versa, but we observed predominant binding of syntaxin-1 to Munc18-1 in both cases (Supplementary Fig. 7), likely because of the relatively long time scale of these experiments (h). We turned to liposome co-floatation assays, which can separate the liposomes from unbound materials in the minute time scale. We note that, because of technical difficulties, there was a natural variability among the results obtained in different experiments performed under the same conditions, but the trends observed in the overall data supported the conclusions from the reconstitution and NMR experiments (Fig. 9, Supplementary Fig. 8a–d). In co-floatation assays where S-liposomes were first incubated with 3 μM αSNAP and different concentrations of Munc18-1 were added before centrifugation, we observed robust αSNAP binding and little Munc18-1 binding, whereas the opposite was observed when the S-liposomes were incubated first with 3 μM Munc18-1 and then different concentrations of αSNAP were added (Fig. 9a, b). In additional experiments, we pre-incubated S-liposomes with 3 μM αSNAP, we added 3 μM Munc18-1 and we incubated for variable times before centrifugation. We observed progressively increased binding of Munc18-1 and displacement of αSNAP as the second incubation was prolonged (Fig. 9c; Supplementary Fig. 8c). Thus, Munc18-1 binds to liposome-anchored syntaxin-1 tighter than αSNAP, but αSNAP binds faster and displacement of αSNAP by Munc18-1 is slow. This conclusion was further confirmed with a gel filtration assay (Fig. 9e–g). When we mixed S-liposomes with 3 μM αSNAP and increasing concentrations of Munc18-1 in the presence of NSF, we observed progressively increased binding of Munc18-1 and decreased binding of αSNAP without the need for long incubations (Fig. 9d). Hence, disassembly of the syntaxin-1-αSNAP complex by NSF accelerates the displacement of αSNAP by Munc18-1.

**αSNAP inhibits the fusion activity of *trans*-SNARE complexes.** Some evidence suggested that αSNAP inhibits liposome fusion by hindering C-terminal *trans*-SNARE complex zippering rather than preventing SNARE-mediated liposome docking[46]. Conversely, yeast Sec17 was shown to promote fusion of liposomes bridged by *trans*-SNARE complexes[40–42]. To directly test whether αSNAP inhibits fusion caused by pre-formed *trans*-SNARE complexes and examine whether such inhibition can be overcome by Munc18-1, M13C$_1$C$_2$BMUNC$_2$C, and/or NSF, we developed an assay that monitors *trans*-SNARE complex formation and lipid mixing between V- and T-liposomes. Synaptobrevin and

syntaxin-1 were labeled with FRET donor and acceptor probes, respectively, and V-liposomes contained 3.5% 1,1′-dioctadecyl-3,3,3′,3′-tetramethylindodicarbocyanine perchlorate (DiD)-labeled lipids, such that we could monitor *trans*-SNARE complex formation through FRET and lipid mixing from de-quenching of DiD fluorescence (Fig. 10a). The V-liposomes were prepared with a low synaptobrevin-to-lipid ratio (1:4000) to avoid formation of large numbers of *trans*-SNARE complexes between pairs of liposomes, which can confound the interpretation of the data, and the T-liposomes (1:800 P:L ratio) were added in excess to maximize the percentage of synaptobrevin molecules involved in complex formation[36].

Incubation of the V- and T-liposomes for 24 h at 4 °C (to prevent fusion) led to a robust decrease in donor fluorescence (Fig. 10b, red vs. black traces) that revealed efficient *trans*-SNARE complex assembly, as only half of the synaptobrevin molecules are expected to be on the outside of the liposomes. The addition of αSNAP did not alter the fluorescence spectrum, but a marked recovery of donor fluorescence was caused by addition of αSNAP together with NSF (Fig. 10b, blue and orange traces), indicating that NSF and αSNAP disassemble *trans*-SNARE complexes albeit not completely, as observed previously[36,38], but αSNAP alone does not dissociate them. We observed robust lipid mixing when we brought samples with the pre-formed *trans*-SNARE complexes to 37 °C, but not when we mixed fresh samples of V- and T-liposomes (Fig. 10c, red and black traces), showing that lipid mixing is mediated by the *trans*-SNARE complexes that were formed during pre-incubation. Importantly, no lipid mixing was observed when αSNAP alone or αSNAP plus NSF were added to pre-formed *trans*-SNARE complexes before bringing them to 37 °C (Fig. 10c, blue and orange traces), showing that αSNAP by itself can abrogate lipid mixing. This activity required binding of αSNAP to the SNAREs and membranes, as the αSNAP KE and FS mutants did not inhibit lipid mixing (Fig. 10d). A titration with WT αSNAP showed that inhibition of lipid mixing occurred with high cooperativity and an EC$_{50}$ (half-maximal effective concentration) of 170 nM (Fig. 10e; Supplementary Fig. 9). The addition of Munc18-1 and M13C$_1$C$_2$BMUNC$_2$C to pre-formed *trans*-SNARE complexes with or without Ca$^{2+}$ before raising the temperature to 37 °C did not substantially alter the lipid mixing induced by the *trans*-SNARE complexes alone (Fig. 10f, blue and pink traces) or the inhibition caused by αSNAP (Fig. 10f, black and orange traces). Slow but efficient lipid mixing was observed when Munc18-1, M13C$_1$C$_2$BMUNC$_2$C, and Ca$^{2+}$ were added to the pre-formed *trans*-SNARE complexes together with NSF and αSNAP (Fig. 10f, gray trace), while no lipid mixing was observed in analogous experiments where Munc18-1 or Ca$^{2+}$ was omitted (Fig. 10f, cyan and red traces). Note that observation of lipid mixing is not sufficient to demonstrate liposome fusion, but the absence of lipid mixing does show that no fusion occurs. Hence, these results show that αSNAP hinders the ability of neuronal *trans*-SNARE complexes to induce membrane fusion. The resulting arrested state (Fig. 1, state 3) can be disassembled by NSF, enabling the Munc18-1-Munc13-1-dependent pathway of membrane fusion.

## Discussion

Extensive studies have provided a wealth of information on the machinery that governs neurotransmitter release, leading to a model whereby Munc18-1 and Munc13s orchestrate *trans*-SNARE complex assembly in an environment that favors SNARE complex disassembly by NSF-SNAPs[3]. This pathway enables a wide range of pre-synaptic plasticity processes that depend on Munc13s and associated proteins[27], and that underlie diverse forms of information processing in the brain[2].

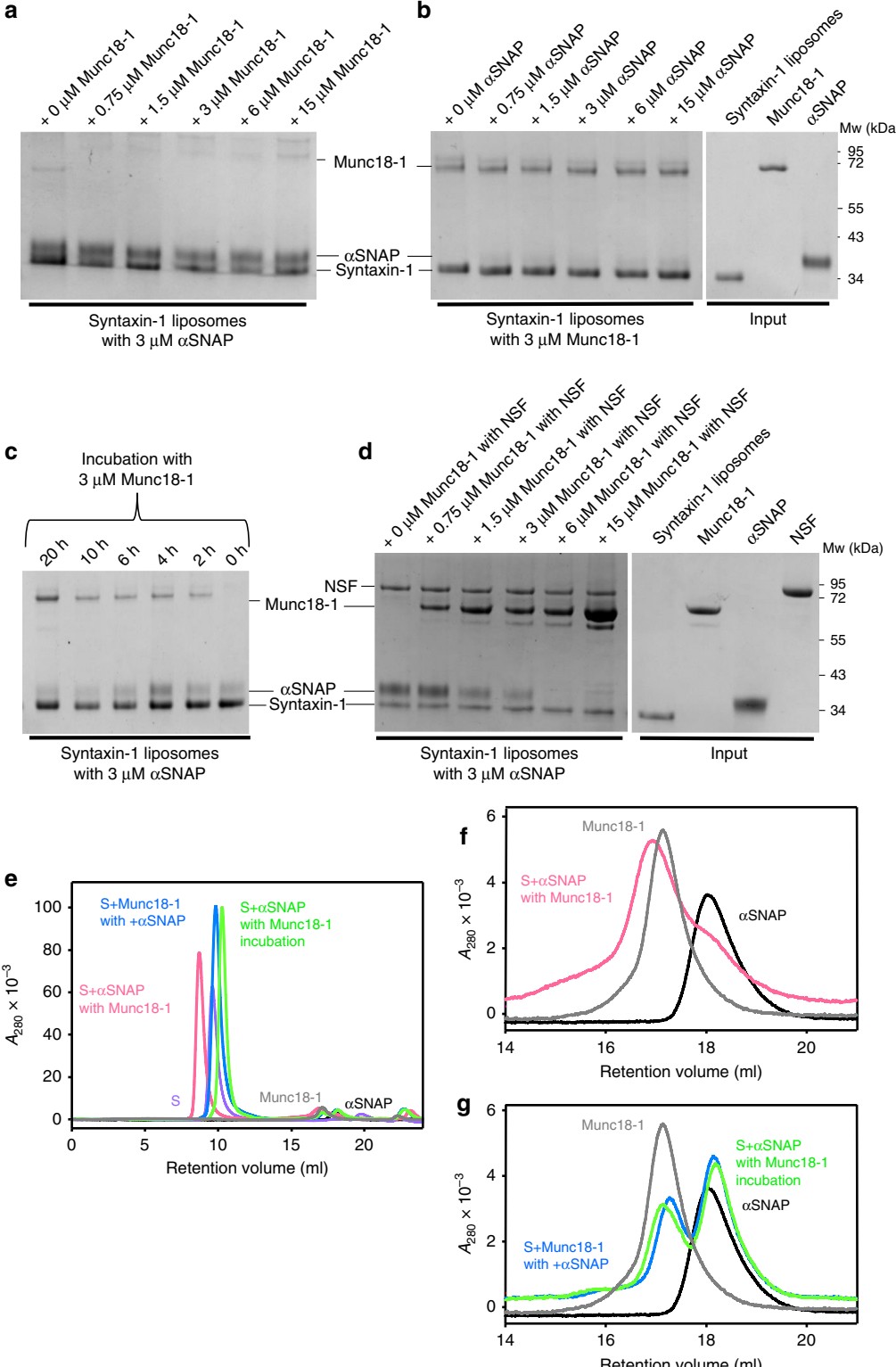

However, it was unclear how this pathway is selected over other SNARE-dependent fusion mechanisms, and key questions remained about the interplay between the core components of the release machinery, including the different states of syntaxin-1. Particularly important was to better understand the NSF-independent functions of SNAPs and how their inhibitory activity is overcome. Our results now show that αSNAP inhibits liposome fusion by multiple mechanisms, in agreement with previous studies[44,46]. Importantly, we now show that αSNAP

inhibition can be overcome only when Munc18-1 binds to closed syntaxin-1, which leads to *trans*-SNARE complex assembly with the crucial assistance of Munc13-1. Together with previous results, our data help to integrate our knowledge on the core neurotransmitter release machinery into a coherent model whereby syntaxin-1 can exist in different states, some of which could potentially lead to fusion, and αSNAP acts as a key inhibitor of non-regulated fusion pathways (Fig. 1), thus playing an indirect but key role in ensuring that neurotransmitter

**Fig. 9** Munc18-1 and αSNAP compete for binding to liposome-anchored syntaxin-1. **a–d** Binding of Munc18-1 and αSNAP to S-liposomes was analyzed using liposome co-floatation assays and SDS-PAGE followed by Coomassie blue staining. In **a**, S-liposomes were pre-incubated for 1 h with 3 μM αSNAP and different concentrations of Munc18-1 were added before co-floatation. In **b**, S-liposomes were pre-incubated for 1 h with 3 μM Munc18-1 and different concentrations of αSNAP were added before co-floatation. In **c**, S-liposomes were pre-incubated for 1 h with 3 μM αSNAP, 3 μM Munc18-1 was then added, and the samples were incubated for the indicated times before co-floatation. In **d**, S-liposomes were pre-incubated for 1 h with 3 μM αSNAP, and 2.4 μM NSF, together with different concentrations of Munc18-1, were added before co-floatation. The strong band of Munc18-1 in the lane corresponding to 15 μM Munc18-1 may arise from aggregation and/or weak binding to the liposomes. Additional co-flotation assays are shown in Supplementary Fig. 8a–d. **e–g** Gel filtration analysis of the competition between Munc18-1 and αSNAP for syntaxin-1 binding. **e** shows the entire chromatograms obtained after injecting the following samples: Munc18-1 alone (gray trace), αSNAP alone (black trace), syntaxin-1 liposomes alone (purple trace), syntaxin-1-liposomes that were incubated with αSNAP and mixed with Munc18-1 before injection (pink trace), syntaxin-1-liposomes that were incubated first with αSNAP and, after the addition of Munc18-1, were further incubated for 12 h (green trace), and syntaxin-1-liposomes that were incubated with Munc18-1 and mixed with αSNAP before injection (blue trace). The liposomes and bound proteins elute near the void volume. Munc18-1 and αSNAP eluting at their characteristic volumes (ca. 17 and 18 ml, respectively) did not bind to the liposomes (see the expansions in **f**, **g**). Munc18-1 displaced much of the αSNAP bound to the syntaxin-1-liposomes if incubated for 12 h, but not if added before injection. When added first, Munc18-1 dominated over αSNAP for binding to the syntaxin-1-liposomes. Source data are provided as a Source Data file

release occurs only at the active zone via the exquisitely regulated Munc18-1-Munc13 pathway.

Previous studies of secretory granule and acrosomal exocytosis, and of lipid mixing using reconstituted proteoliposomes, revealed NSF-independent inhibitory functions of αSNAP[43–46], but yielded diverse conclusions that were difficult to incorporate into a unified model (see Introduction). The data presented here are consistent with many of their results and show that αSNAP can potently inhibit membrane fusion by three mechanisms: (i) binding to isolated syntaxin-1, thus precluding binding of syntaxin-1 to Munc18-1 and SNAP-25 (Figs. 2, 5, 7–9); (ii) binding to syntaxin-1-SNAP-25 heterodimers, thus hindering *trans*-SNARE complex assembly (Figs. 2, 5, 6); and (iii) binding to *trans*-SNARE complexes, directly preventing membrane fusion (Fig. 10). These results can be rationalized based on the cryo-electron microscopy structure of the 20S complex formed by NSF, αSNAP, and the SNARE four-helix bundle[53], which shows how four αSNAP molecules bind around much of the surface of the four-helix bundle through largely electrostatic interactions. Given the promiscuity of these interactions and the overall negative charge of the SNARE motifs, it is natural that αSNAP can bind in similar modes to tetrameric structures formed by 2:1 syntaxin-1-SNAP-25 heterodimers[64] or by four syntaxin-1 SNARE motifs, thus inhibiting interactions with these four-helix bundles. While formation of such syntaxin-1 tetramers on liposomes remains to be fully demonstrated, this notion is supported by our NMR data showing that syntaxin-1 adopts an open conformation on S-liposomes and binds tightly to αSNAP, but not to SNAP-25 in this state, obstructing the very strong closed syntaxin-1–Munc18-1 interaction (Figs. 8, 9; Supplementary Fig. 5). Our data agree with results indicating that αSNAP inhibits fusion by binding to syntaxin-1[44,45] or by impeding full *trans*-SNARE complex zippering[46], but the inhibition of fusion between T- and V-liposomes that we observed is much stronger than previously observed[44]. It is plausible that this discrepancy arises because the latter study used a syntaxin-1 fragment lacking the N-terminal region, whereas we used full-length syntaxin-1, or because we purified syntaxin-1 using dodecylphosphocholine (DPC), which prevents formation of large syntaxin-1 aggregates in contrast to other detergents[59].

The inhibitory activities of αSNAP involving binding to syntaxin-1 or syntaxin-1-heterodimers correlate with the finding that Sec17 inhibits early stages of yeast vacuolar fusion[39,40]. However, the strong inhibition of liposome fusion by pre-formed *trans*-SNARE complexes (Fig. 10) represents a striking difference with respect to ample data showing that Sec17 can strongly stimulate the ability of *trans*-SNARE complexes to mediate fusion[40–42]. Note that the activity of Sec17 as an inhibitor or

activator depends on its concentration and the lipid composition of the liposomes[65]. Hence, further research will be required to assess whether there is a true functional divergence between αSNAP and Sec17. However, such divergence would not be surprising considering that other key properties of the neurotransmitter release machinery are not generally conserved. These include the syntaxin-1 closed conformation[12], which is not adopted by Vam3[28], and its interaction with Munc18-1[12,13], which was not observed for the yeast exocytotic homologs[66]. It is tempting to speculate that including a protein such as Sec17, which can bind to the SNARE complex to facilitate fusion and help to disassemble the complex after fusion, represents an economic means to couple constitutive membrane traffic with SNARE recycling. However, this design might limit the range of possible mechanisms that regulate fusion. It appears that, to meet the exquisite temporal and regulatory requirements of synaptic vesicle exocytosis, billions of years of evolution led to unique features of the core components of the synaptic vesicle fusion machinery and to specialized factors such as synaptotagmin-1 and complexins, which bind to the SNARE four-helix bundle [reviewed in ref. 3,5]. Because these interactions are incompatible with αSNAP binding, the macromolecular assembly that causes synaptic vesicle fusion is not expected to include αSNAP bound to the SNAREs. Indeed, complexin-1 was shown to compete with αSNAP for SNARE complex binding[67], and one of the roles of synaptotagmin-1 and complexin-1 may be to prevent binding of αSNAP to the SNAREs both to circumvent the inhibitory activity of αSNAP and to protect against disassembly of *trans*-SNARE complexes by NSF-αSNAP, a role that is likely played also by Munc18-1 and Munc13-1[36].

The inhibitory activity of αSNAP involving binding to *trans*-SNARE complexes may be a last resort to preclude constitutive fusion, which would have disastrous consequences for synapse function, whereas binding of αSNAP to syntaxin-1-SNAP-25 heterodimers can prevent fusion at earlier stages by hindering *trans*-SNARE complex assembly (Fig. 6). NSF likely helps also to impede constitutive fusion by disassembling syntaxin-1-SNAP-25 heterodimers[32] and *trans*-SNARE complexes[36,38] with the assistance of αSNAP. However, some *trans*-SNARE complexes are resistant to disassembly by NSF-αSNAP[36,38] and syntaxin-1 and SNAP-25 are highly abundant[33], co-localizing within clusters that they form on the plasma membrane[34]. Hence, the existence of a population of syntaxin-1-SNAP-25 heterodimers seems unavoidable, and the binding of αSNAP to such heterodimers or to potential *trans*-SNARE complexes that they might form with synaptobrevin may be a first, fast, crucial stop-gap to ensure that fusion does not occur constitutively. NSF can later disassemble the resulting complexes to "reset" the system and favor the

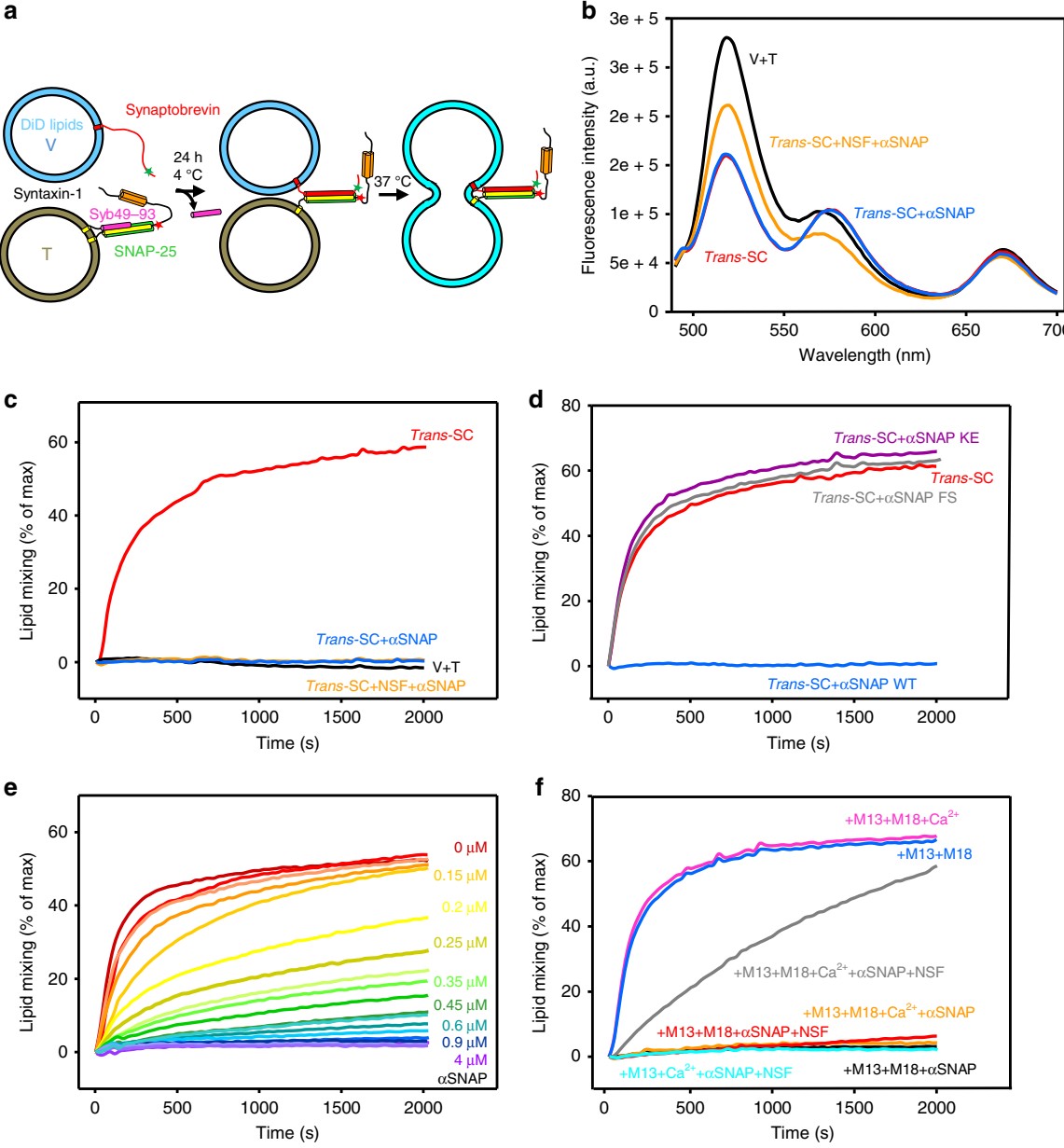

**Fig. 10** αSNAP potently inhibits fusion mediated by pre-formed *trans*-SNARE complexes. **a** Diagram illustrating the experimental design. The assays monitored the development of FRET between V-liposomes containing Alex488-synaptobrevin and T-liposomes containing tetramethylrhodamine (TMR)-syntaxin-1 and WT SNAP-25 upon *trans*-SNARE complex assembly. The design is analogous to that of Fig. 6a, but the use of WT SNAP-25 allows liposome fusion. The V-liposomes contained DiD-labeled lipids to monitor lipid mixing from de-quenching of the DiD fluorescence. V- and T-liposome samples were incubated with Syb49–93 for 24 h at 4 °C to promote *trans*-SNARE complex assembly without liposome fusion, and the temperature was raised to 37 °C after adding various factors to allow lipid mixing. **b** Fluorescence emission spectra (468 nm excitation) of a mixture of V-liposomes containing Alexa-488-synaptobrevin and T-liposomes containing TMR-syntaxin-1 and WT SNAP-25 (1:4 V- to T-liposome ratio) that had been incubated for 24 h with Syb49–93 at 4 °C (red trace) and of the same sample after adding 2 μM αSNAP (blue trace) or 2 μM αSNAP plus 0.4 μM NSF, 2 mM ATP, and 2.5 mM Mg$^{2+}$ (orange trace). The black curve shows a control spectrum obtained by adding spectra acquired separately for V- and T-liposomes. **c–f** Lipid mixing assays performed as summarized in **a**. De-quenching of DiD fluorescence was monitored after adding the indicated reagents to the *trans*-SNARE complexes pre-formed at 4 °C and raising the temperature to 37 °C. **c** Assays performed with no additions (red trace) or adding 2 μM αSNAP without (blue trace) or with 0.4 μM NSF (orange trace). The black trace shows an experiment were the V- and T-liposomes were mixed at 37 °C without pre-incubation at 4 °C. **d** Assays analogous to those in **b** with the addition of 2 μM WT αSNAP or the αSNAP FS or KE mutants. **e** Assays analogous to those of **c** with the addition of different concentrations of αSNAP. **f** Assays analogous to those of **c** with the addition of different combinations of 1 μM Munc18-1 (M18), 0.3 μM M13C$_1$C$_2$BMUNC$_2$C (M13), 0.4 μM NSF, 2 μM αSNAP, and 0.5 mM Ca$^{2+}$. Source data are provided as a Source Data file

Munc18-1-Munc13-1 pathway (Figs. 4, 10f). This re-setting process appears to be slow after αSNAP binds to pre-formed *trans*-SNARE complexes (Fig. 10f, gray trace). However, this slow speed may arise from formation of too many *trans*-SNARE complexes and/or of extended membrane–membrane interfaces[38]

during the long incubation at low temperature, causing steric hindrance that impairs NSF access. In contrast, NSF quickly restored fusion in assays where αSNAP arrested fusion between V- and T-liposomes in the presence of Munc18-1 and M13C$_1$C$_2$BMUNC$_2$C (Fig. 4c, d).

Importantly, our data strongly suggest that the only mechanism to avoid the inhibitory activities of αSNAP is for Munc18-1 to bind to closed syntaxin-1, which initiates the proper pathway to synaptic vesicle fusion that requires Munc13-1. This pathway is also hindered by αSNAP because of its interaction with isolated syntaxin-1 (Fig. 7), but Munc18-1 can displace αSNAP directly in a slow time scale or faster when NSF disassembles the αSNAP-syntaxin-1 complex and renders closed syntaxin-1[54] (Fig. 9c, d). The total abrogation of neurotransmitter release observed in the absence of Munc18-1[23] showed the essential nature of this protein. This crucial importance is further emphasized by the fact that the total abrogation of release observed in the absence of Munc13s can be partially rescued by mutations in syntaxin-1 or Munc18-1[20,29], or by inactivating NSF with NEM[37], but no means to rescue release in Munc18-1 KO neurons have been described, and NEM failed to elicit such rescue[37]. The central roles of Munc18-1 in templating SNARE complex assembly[14–17] and organizing assembly in an NSF-αSNAP resistant manner[11,36] most likely underlie in part the dramatic Munc18-1 KO phenotype, but it is unclear whether these functions alone can explain the absolute absence of release without Munc18-1, given the presence of syntaxin-1-SNAP-25 heterodimers in the plasma membrane[34] that could mediate fusion[35]. Thus, the most crucial reason for the essential nature of Munc18-1 may be the need to overcome the inhibitory activities of αSNAP by binding to closed syntaxin-1. Note that, in yeast vacuolar fusion, the HOPS complex also overcomes an early inhibitory activity of Sec17[40], but it seems likely that the underlying mechanism is different, as Vam3 does not adopt a closed conformation[28].

The existence of multiple states of syntaxin-1 that interconvert slowly has been recognized for a long time [reviewed in ref. [27]]. These features complicate the interpretation of reconstitution assays and also of genetic studies of the release machinery, as the different states of syntaxin-1 are likely to exist in vivo given the promiscuity of its SNARE motif. Genetic studies of αSNAP (e.g., ref. [68]) are further complicated because of its general function in SNARE complex disassembly, in addition to its role in inhibiting exocytosis. Our results help to "disentangle" the complexity of this system and to integrate our knowledge into a plausible model for the interplay between the core components of the release apparatus (Fig. 1) that can be used as a framework for future studies of the mechanism of neurotransmitter release. Clearly, additional studies in neurons will be required to test these ideas, but it is rewarding that this model provides natural explanations for a large amount of available physiological data.

## Methods

**Protein expression and purification.** Bacterial expression and purification of full-length rat syntaxin-1A, the cytoplasmic fragment of rat syntaxin-1A (residues 2–253), a cysteine-free variant of full-length rat SNAP-25a, a full-length rat synaptobrevin-2, rat synaptobrevin-2(49–93), full-length rat Munc18-1, rat synaptotagmin-1 57–421 (C74S, C75A, C77S, C79I, C82L, C277S), full-length rat complexin-1, full-length *Cricetulus griseus* WT or V155M mutant NSF, full-length *Bos Taurus* αSNAP, and a rat Munc13-1 C₁C₂BMUNC₂C fragment (residues 529–1725, Δ1408–1452) were described previously[11,12,18,36,54,69,70]. Briefly, all proteins were expressed in *Escherichia coli* BL21(DE3) cells in Luria-Bertani media, expect synaptotagmin-1 that was expressed in Terrific Broth media, and isotopically labeled proteins, which were expressed in minimal media with $^{15}NH_4Cl$ (Sigma; 1 g/L of culture) as the sole nitrogen source for uniform $^{15}N$ labeling. For $^2H$-I-$^{13}CH_3$ labeling, minimal media was made in $D_2O$ with $^2H$-glucose (3 g/L of culture) as the main carbon source, and [3,3-$^2H$] $^{13}C$-methyl α-ketobutyric acid (Sigma; 80 g/L of culture) was added to the cell cultures 30 min prior to induction. Isotopically labeled proteins were purified as unlabeled proteins.

Syntaxin-1A was expressed overnight at 25 °C upon induction with 0.4 mM isopropyl-β-D-thiogalactoside (IPTG). Cell pellets were re-suspended in 20 mM HEPES, pH 7.4, 500 mM NaCl, 8 mM imidazole, 1 mM TCEP (tris(2-carboxyethyl) phosphine), and upon cell lysis, protein purification was performed using HisPur Ni-NTA resin (Thermo Fisher) in 20 mM Tris, pH 7.4, 500 mM NaCl, 8 mM imidazole, 2% Triton X-100 (v/v), 6 M urea, followed by elution in 20 mM Tris, pH 7.4, 500 mM NaCl, 400 mM imidazole, and 0.1% DPC. The polyhistidine tag was

removed using thrombin protease, followed by size-exclusion chromatography on a Superdex 200 column (GE 10/300) equilibrated in 20 mM Tris, pH 7.4, 125 mM NaCl, 1 mM TCEP, and 0.2% DPC. The buffer contained DPC to prevent aggregation[59].

Expression of syntaxin-1A 2–253 was induced with 0.4 mM IPTG and expressed overnight at 25 °C. Upon cell lysis, purification was done using Glutathione Sepharose 4B resin (GE) in phosphate-buffered saline (PBS), PBS with 1% Triton X-100 (v/v), and PBS with 1 M NaCl. The GST-tag was cleaved by thrombin protease and the eluted protein was further purified by anion exchange chromatography on a HiTrap Q column (GE) in 25 mM Tris, pH 7.4, and 1 mM TCEP using a linear gradient from 0 to 1 M NaCl.

Cysteine-free SNAP-25a was expressed overnight at 25 °C upon induction with 0.4 mM IPTG. Upon lysis, protein purification was performed using HisPur Ni-NTA resin (Thermo Fisher) in 50 mM Tris, pH 8.0, 500 mM NaCl, 20 mM imidazole, and 1% Triton X-100 (v/v). The His₆-tag was cleaved by thrombin protease, and the protein was purified by size-exclusion chromatography on a Superdex 75 column (GE 16/60) in 50 mM Tris, pH 8.0, and 150 mM NaCl.

Full-length synaptobrevin-2 was expressed overnight at 25 °C upon induction with 0.4 mM IPTG. Cells were re-suspended in PBS buffer containing 1% Triton X-100 (v/v). Purification was done using Glutathione Sepharose 4B resin (GE) at 4 °C. The bound proteins were treated with PBS with 1% Triton X-100 (v/v), followed by the addition of thrombin to cleave the GST-tag. The protein was further purified by cation exchange chromatography on a HiTrap S column (GE) in 25 mM NaAc, pH 5.5, 1 mM TCEP, and 1% β-octyl glucoside (β-OG) (w/v) using a linear gradient from 0 to 1 M NaCl.

Synaptobrevin-2(49–93) was induced with 0.4 mM IPTG and expressed overnight at 23 °C. Purification was done using Glutathione Sepharose 4B resin (GE), followed by cleavage of the GST-tag. The final purification step was size-exclusion chromatography on a Superdex 75 column (GE 16/60) equilibrated in 20 mM Tris, pH 7.4, and 125 mM NaCl.

Expression of full-length Munc18-1 was induced with 0.4 mM IPTG and continued overnight at 20 °C. Upon cell lysis and centrifugation, the supernatant was loaded on Glutathione Sepharose 4B resin (GE) at 4 °C and the bound proteins were washed with PBS, PBS with 1% Triton X-100 (v/v), and PBS with 1 M NaCl. The GST-tag was cleaved from the protein with thrombin, followed by immediate size-exclusion chromatography using a Superdex 200 column (GE 16/60) in a buffer containing 20 mM Tris, pH 7.4, 200 mM KCl, and 1 mM TCEP.

The plasmid encoding rat synaptotagmin-1(57–421) was a kind gift from Thomas Söllner. The protein was expressed in Terrific Broth media overnight at 20 °C upon induction with 0.4 mM ITPG. Cell pellets were re-suspended in 25 mM HEPES, pH 7.4, 600 mM KCl, and lysed using high-pressure homogenization (Avestin). Upon centrifugation (48,298 × g, 30 min, 4 °C), the soluble fraction of the cell lysate was harvested and 1.5% Triton X-100 (v/v) was slowly added and stirred at 4 °C for 2 h. Upon further centrifugation, the supernatant was incubated with PurHis Ni-NTA resin (Thermo Fisher) at 4 °C for 2 h. The resin was washed with wash buffer 25 mM HEPES, pH 7.4, 600 mM KCl, 10 mM imidazole, and 1% β-OG (w/v). Upon elution with 25 mM HEPES, pH 7.4, 600 mM KCl, 250 mM imidazole, 1% β-OG (w/v), the protein was further purified by size-exclusion chromatography on a Superdex 200 column (GE 16/60) in 25 mM HEPES, pH 7.4, 600 mM KCl, and 1% β-OG (w/v).

Expression of full-length complexin-1 was induced with 0.5 mM IPTG and lasted 4 h at 37 °C. Purification was done using HisPur Ni-NTA resin (Thermo Fisher) and followed by TEV cleavage of the His₆-tag. The protein was further purified by size-exclusion chromatography on a Superdex 75 column (GE 16/60) in running buffer 20 mM Tris, pH 7.4, 125 mM NaCl, and 1 mM TCEP.

Expression of α-SNAP was inducted by the addition of 0.4 mM ITPG and continued overnight at 25 °C. Protein purification was performed using Glutathione Sepharose 4B resin (GE) by washing bound proteins with PBS, PBS with 1% Triton X-100 (v/v), and PBS with 1 M NaCl. Upon GST-tag cleavage in the presence of thrombin, the protein was purified by size-exclusion chromatography using a Superdex 75 column (GE 16/60) in 20 mM Tris, pH 7.4, 150 mM KCl, and 1 mM TCEP.

Full-length NSF was induced with 0.4 mM IPTG and expressed overnight at 20 °C. Purification was performed using HisPur Ni-NTA resin (Thermo Fisher), followed by size-exclusion chromatography of hexameric NSF on a Superdex S200 column (GE 16/60) in 50 mM Tris, pH 8.0, 100 mM NaCl, 1 mM ATP, 1 mM EDTA, 1 mM dithiothreitol, and 10% glycerol (v/v). Removal of the His₆-tag and monomerization of NSF was done using TEV protease and apyrase, respectively, while dialyzing with nucleotide-free buffer for 36 h. To separate the hexameric form of NSF from the monomeric, three rounds of size-exclusion chromatography on a Superdex S200 column (GE 16/60) in 50 mM NaPi, pH 8.0, 100 mM NaCl, and 0.5 mM TCEP were performed by re-injecting fractions with hexameric NSF. Final reassembly of monomers and gel filtration chromatography of reassembled hexameric NSF were done using a Superdex S200 column (GE 16/60) in 50 mM Tris, pH 8.0, 100 mM NaCl, 1 mM ATP, 1 mM EDTA, 1 mM TCEP, and 10% glycerol (v/v). For experiments conducted with non-hydrozable ATP analog, purification of NSF was performed in the presence of ATPγS instead of ATP.

Expression of a rat Munc13-1 C1C2BMUNC2C fragment (residues 529–1725, Δ1408–1452) was induced with 0.5 mM IPTG and performed overnight at 16 °C. Upon cell lysis and centrifugation, the supernatant was loaded onto HisPur Ni-NTA resin (Thermo Fisher) and washed with 50 mM Tris, pH 8, 10 mM imidazole,

750 mM NaCl, 1 mM TCEP, and 10% glycerol (v/v). The protein was eluted with 50 mM Tris, pH 8, 250 mM NaCl, 1 mM TCEP, 10% glycerol (v/v), 500 mM imidazole and dialyzed overnight at 4 °C in 50 mM Tris, pH 8, 250 mM NaCl, 1 mM TCEP, 2.5 mM CaCl$_2$, 10% glycerol (v/v) in the presence of thrombin. The protein was further purified by anion exchange chromatography on a HiTrap Q column (GE) in 20 mM Tris, pH 8.0, 1 mM TCEP, and 10% glycerol (v/v) using a linear gradient from 0 to 1 M NaCl.

All the following mutants were generated using the QuickChange site-directed mutagenesis and custom-designed primers (listed in Supplementary Table 1): syntaxin-1A 1–288 Δ21–189 (ΔH$_{abc}$), syntaxin-1A S186C in cysteine-free full-length syntaxin-1A (C145A, C271A, C272A), synaptobrevin-2 L26C mutation in cysteine-free full-length synaptobrevin-2 (C103A), SNAP-25a M71D,L78D (ML) in a cysteine-free construct, αSNAP K122E,K163E (αSNAP KE), αSNAP F27S,F28S (αSNAP FS), and αSNAP with a C-terminal His$_6$-linker (LRLPETGSGSHHHHHHAA). All mutant proteins were purified as the unmodified constructs.

**Simultaneous lipid mixing and content mixing assays.** Assay that simulta-neously measures lipid and content mixing was performed as described in detail in ref. [18]. Briefly, V-liposomes containing full-length synaptobrevin-2 (protein-to-lipid ratio 1:500) were made with 39% POPC (1-palmitoyl, 2-oleoyl phosphati-dylcholine), 19% DOPS (1,2-dioleoyl-sn-glycero-3-phospho-L-serine), 19% POPE (1-palmitoyl-2-oleoyl-sn-phosphatidylethanolamine), 20% cholesterol, 1.5% NBD-PE (N-(7-nitrobenz-2-oxa-1,3-diazol-4-yl)-1,2-dihexadecanoyl-sn-glycero-3-phos-phoethanolamine, triethylammonium salt), and 1.5% Marina Blue DHPE (1,2-dihexadecanoyl-sn-glycero-3-phosphoethanolamine). VS-liposomes with synapto-tagmin-1(57–421) and full-length synaptobrevin (synaptotagmin-1:synaptobrevin:lipid ratio 1:2:1,000) contained 40% POPC, 6.8% DOPS, 30.2% POPE, 20% cho-lesterol, 1.5% NBD-PE, and 1.5% Marina Blue DHPE. T-liposomes containing syntaxin-1 (WT or ΔH$_{abc}$) and SNAP-25 (syntaxin-1:lipid ratio 1:800) and S-liposomes with full-length syntaxin-1A (protein-to-lipid ratio 1:800) were made with 38% POPC, 18% DOPS, 20% POPE, 20% cholesterol, 2% PIP2 (phosphati-dylinositol 4,5-biphosphate), and 2% DAG (diacylglycerol). Dried lipid films were re-suspended in 25 mM HEPES, pH 7.4, 150 mM KCl, 1 mM TCEP, 10% glycerol (v/v), and 2% β-OG. Lipid solutions were then mixed with the respective proteins and with 4 μM phycoerythrin-biotin for T-liposomes or with 8 μM Cy5-streptavidin for V- or VS-liposomes in 25 mM HEPES, pH 7.4, 150 mM KCl, 1 mM TCEP, and 10% glycerol (v/v). Proteoliposomes were prepared by detergent removal using dialysis with 2 g/L Amberlite XAD-2 beads (Sigma) three times at 4 °C and subsequent co-floatation on a three-layer histodenz gradient (35%, 25%, and 0%), and harvested from the topmost layer. Lipid mixing was measured by monitoring fluorescence de-quenching of Marina Blue-labeled lipids (excitation at 370 nm, emission at 465 nm) and content mixing was measured from the devel-opment of FRET between Cy5-Streptavidin trapped in V- or VS-liposomes and phycoerythrin-biotin trapped in T-liposomes (excitation at 565 nm, emission at 670 nm). Each reaction was prepared in a total volume of 200 μl with V- or VS-liposomes (0.125 mM total lipid), T-liposomes or S-liposomes (0.25 mM total lipid), 2.5 mM MgCl$_2$, 2 mM ATP, 0.1 mM EGTA, 5 μM streptavidin, and various additions of other components: SNAP-25, NSF purified in ATP or ATPγS con-taining buffer, αSNAP (WT, or FS or KE mutant), Munc13-1 C$_1$C$_2$BMuncC$_2$C, and complexin in different combinations (concentrations indi-cated in the figure legends). At 300 s, CaCl$_2$ (0.6 mM) was added to each reaction mixture. All experiments were repeated at least three times with a given prepara-tion and the results were verified in multiple experiments performed with different preparations. All assays were performed at 30 °C using a PTI Quantamaster 400 spectrofluorometer (T-format) equipped with a rapid Peltier temperature-controlled four-position sample holder (all slits set to 1 mm). Lipid and content mixing were normalized to the maximum signals obtained by the addition of 1% β-OG at the end of each (for lipid mixing) or to controls acquired without streptavidin in the presence of 1% β-OG to measure the maximal Cy5 fluorescence (content mixing).

**Trans-SNARE complex formation assay.** To monitor *trans*-SNARE complex as a function of time without interference from membrane fusion, we used a SNAP-25a mutant bearing two single residue substitutions M71D,L78D (SNAP-25m)[36]. Single cysteine variants of syntaxin-1A (S186C) and synaptobrevin-2 (L26C) were, respectively, labeled with tetramethylrhodamine (TMR) and Alexa-488, as descri-bed[36]. Proteoliposomes were prepared similarly to those used for lipid mixing and content mixing fusion assays with several exceptions. V-liposomes with full-length synaptobrevin-2 L26C-Alexa-488 (protein-to-lipid ratio 1:10,000) contained 42% POPC, 19% DOPS, 19% POPE, and 20% cholesterol. The lipid composition of T-liposomes with full-length syntaxin-1A S186C-TMR and full-length SNAP-25m consisted of 38% POPC, 18% DOPS, 20% POPE, 20% cholesterol, 2% PIP2, and 2% DAG (syntaxin-to-lipid ratio 1:800). The proteoliposomes were prepared in 25 mM HEPES, pH 7.4, 150 mM KCl, and 1 mM TCEP. *Trans*-SNARE complex formation was measured by the development of FRET between Alexa-488-synaptobrevin on V-liposomes (0.0625 mM total lipid) and TMR-syntaxin-1A on T-liposomes (0.25 mM total lipid) at 37 °C using a PTI Quantamaster 400 spectrofluorometer (T-format) equipped with a rapid Peltier temperature controlled four-position sample holder, with all slits set to 1.25 mm. The fluorescence signal of V-liposomes

at 518 nm (excitation at 468 nm) was recorded to monitor the Alexa-488 donor fluorescence intensity over time upon mixing with acceptor T-liposomes, 10 μM synaptobrevin-2(49–93), 0.3 μM Munc13-1 C$_1$C$_2$BMuncC$_2$C in the absence or presence of 2 μM αSNAP. A GG495 longpass filter (Edmund optics) was used to filter scattered light.

**NMR spectroscopy.** All NMR spectra were acquired at 25 °C on an Agilent DD2 spectrometer operating at 800 MHz. $^1$H-$^{13}$C HMQC spectra were acquired with samples containing: 15 μM $^2$H-I-$^{13}$CH$_3$-syntaxin-1(2–253) alone, bound to 15 μM Munc18-1 or incorporated into a SNARE complex formed with the SNARE motifs of synaptobrevin and SNAP-25; S-liposomes containing $^2$H-I-$^{13}$CH$_3$-syn-taxin-1A (12 μM protein) with or without 12 μM Munc18-1 or 26 μM αSNAP; S-liposomes containing $^2$H-I-$^{13}$CH$_3$-syntaxin-1A (7.8 μM protein) with 7.8 μM Munc18-1 and 18 μM αSNAP (incubating first with Munc18-1 and later with αSNAP, or vice versa). All samples were dissolved in 20 mM HEPES, pH 7.4, 125 mM KCl, 1 mM TCEP using D$_2$O as the solvent. $^1$H-$^{15}$N HSQC spectra were acquired with samples containing 9 μM $^{15}$N-SNAP-25 with or without S-liposomes containing $^2$H-I-$^{13}$CH$_3$-syntaxin-1A (11 μM protein) or 11 μM syntaxin-1A 2–253. Samples were dissolved in 20 mM HEPES, pH 7.4, 125 mM KCl, and 1 mM TCEP containing 10% D$_2$O. Total acquisition times were 3 h. Spectra were processed with NMRPipe[71] and analyzed with NMR view[72].

**Liposome co-floatation assays.** S-liposomes with full-length syntaxin-1A were prepared as described above. S-liposomes (0.75 mM total lipid) were either incu-bated for 1 h at room temperature with 3 μM Munc18-1 and then titrated with 0, 0.75, 1.5, 3, 6, and 15 μM αSNAP or incubated for 1 h at room temperature with 3 μM αSNAP containing a linker at the C terminus and then titrated with 0, 0.75, 1.5, 3, 6, and 15 μM Munc18-1 in the absence or presence of 2.4 μM NSF (Fig. 9a, b, d). Alternatively, S-liposomes (0.75 mM total lipid) were incubated with 3 μM αSNAP containing a linker at the C terminus for 1 h at room temperature and then incubated with 3 μM Munc18-1 at 4 °C for various times (Fig. 9c). Analysis of which proteins were bound to the S-liposomes was carried out using a co-flotation assay. Briefly, samples were mixed with an equal volume of 80% Histodenz (w/v) and then transferred to 5 mm by 41-mm centrifuge tubes. Proteoliposomes were then overlaid with 150 μl of 35% and 150 μl of 30% Histodenz (w/v), and then with 30 μl of 25 mM HEPES buffer, pH 7.4, 150 mM KCl, 1 mM TCEP, and 10% gly-cerol (v/v). The gradients were subjected to ultracentrifugation for 4 h at 48,000 r.p. m. in an SW55 rotor (Beckman). The top layer of the gradient (20 μl) was collected and analyzed by sodium dodecyl sulfate-polyacrylamide gel electrophoresis and Coomassie Blue staining. In all experiments, αSNAP contained a His$_6$-linker at the C terminus to allow separation of the syntaxin-1 and αSNAP bands in the gels. Uncropped versions of the gels are shown in the Source Data file.

**Gel filtration binding assay.** S-liposomes with full-length syntaxin-1A were prepared as described above. S-liposomes (2.29 mM total lipid) were either incu-bated for 1 h at room temperature with 1.2 μM αSNAP and then mixed with 1.2 μM Munc18-1, or incubated for 1 h at room temperature with 1.2 μM Munc18 and then mixed with 1.2 μM αSNAP. Alternatively, S-liposomes (2.288 mM total lipid) were incubated with 1.2 μM αSNAP for 1 h at room temperature and then incubated with 1.2 μM Munc18-1 at 4 °C for 12 h. Each sample was prepared in a total volume of 200 μl in 25 mM HEPES, pH 7.4, 150 mM KCl, 1 mM TCEP, and injected into a size-exclusion chromatography column (Superdex 30 Increase 10/300 GL) using a running buffer containing 25 mM HEPES, pH 7.4, 150 mM KCl, and 1 mM TCEP. Isolated Mun18-1 and αSNAP at the same concentrations were also injected to determine their characteristic elution volumes.

**Trans-SNARE complex formation and lipid mixing assays.** To monitor *trans*-SNARE complex formation and lipid mixing simultaneously, we used a cysteine-free variant of full-length SNAP-25a and full-length syntaxin-1A S186C-TMR, which were reconstituted on T-liposomes, and full-length synaptobrevin-2 L26C-Alexa-488 reconstituted on V-liposomes. Proteoliposomes were prepared similarly to that used for the *trans*-SNARE complex formation assay with several exceptions. V-liposomes with synaptobrevin-2 L26C-Alexa-488 (protein-to-lipid ratio 1:4000) contained 38.5% POPC, 19% DOPS, 19% POPE, 20% cholesterol, and 3.5% of a fluorescent probe DiD. The lipid composition and protein-to-lipid ratio of T-liposomes with full-length syntaxin-1A S186C-TMR and full-length SNAP-25 were the same as described above. Pre-formed *trans*-SNARE complexes were prepared by incubating V-liposomes (0.0625 mM total lipids), T-liposomes (0.25 mM total lipid), 2 μM SNAP-25, 10 μM synaptobrevin 49–93, and 0.1 mM EGTA for 24 h at 4 °C, in reaction buffer (25 mM HEPES, pH 7.4, 150 mM KCl, 1 mM TCEP, 2.5 mM MgCl$_2$, and 2 mM ATP). A fluorescence emission scan from 490 to 700 nm (excitation 468 nm) was then collected at 20 °C to measure the amount of *trans*-SNARE complex formed based on the fluorescence signal at 518 nm of the donor V-liposomes. The wavelength scan was also collected for the pre-formed *trans*-SNARE complex with the addition of 2 μM αSNAP. We also added 0.4 μM NSF and 2 μM αSNAP to a pre-formed *trans*-SNARE complex sample to disassemble the complex, incubated for 5 min at 37 °C, and collected a second emission scan. To monitor lipid mixing we initiated the reaction by increasing the temperature to 37 °C and we monitored the de-quenching of the DiD fluorescence at 670 nm

(excitation at 580 nm) of the formed SNARE complex alone or with various additions: 2 μM αSNAP (WT, or KE or FS mutants), 0.4 μM NSF, 0.3 μM Munc13-1 $C_1C_2BMuncC_2C$, 1 μM Munc18-1, and 0.6 mM $CaCl_2$. The fluorescence signal was normalized to the maximum signal induced by the addition of 1% w/v β-OG at the end of the experiments. All experiments were performed on a PTI Quantamaster 400 spectrofluorometer (T-format) equipped with a rapid Peltier temperature-controlled four-position sample holder, with all slits set to 1.25 mm and with a mounted GG495 longpass filter (Edmund optics) to filter scattered light.

**Statistics**. Data from simultaneous lipid mixing and content mixing assays were analyzed from three independent experiments performed with the same proteoliposomes preparations for each reaction condition. Data are expressed as mean ± standard deviation. Statistical comparisons were performed by one-way analysis of variance of the data obtained under different conditions within a set of experiments (Holm–Sidak method, as implemented in Sigma Plot), as shown in the figures and summarized in Supplementary Data 1. A $p$ value in each of group comparison that is smaller than the critical level of each group comparison indicates that there are significant differences between the two tested populations.

**Reporting summary**. Further information on research design is available in the Nature Research Reporting Summary linked to this article.

## Data availability
Data supporting the findings of this manuscript are available from the corresponding author upon reasonable request. A reporting summary for this article is available as a Supplementary Information file. The source data underlying Figs. 2, 3, 4, 5, 6, 7, 9 and 10, and Supplementary Figs. 1, 2, 3, 4, 6, 8 and 9 are provided as a Source Data file.

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

## Acknowledgements

We thank William Wickner and Reinhard Jahn for insightful comments on the manuscript, Minglei Zhao for providing the plasmid to express NSF, and Yun-Zu Pan, Bradley Quade, and Junjie Xu for providing purified proteins. The Agilent DD2 console of the 800 MHz spectrometer used for the research presented here was purchased with a shared instrumentation grants from the NIH (S10OD018027 to J.R.). Eric Prinslow was supported by NIH Training Grant T32 GM008297. This work was supported by grant I-1304 from the Welch Foundation (to J.R.) and by NIH Research Project Award R35 NS097333 (to J.R.).

## Author contributions

All authors participated in the design of the research, as well as in the analysis and interpretation of the data. K.P.S. performed most of the experiments. E.A.P. performed some of the *trans*-SNARE complex assembly assays. J.R. and K.P.S. wrote the paper with input from E.A.P.

## Additional information

**Competing interests:** The authors declare no competing interests.

