## [Peer Review File · Nature Communications]

Reviewers' comments:

Reviewer #1 (Remarks to the Author):

The manuscript by Stepien et al describes novel mechanisms by which the soluble NSF adaptor protein α SNAP inhibits SNARE-dependent fusion in vitro. Using liposome fusion assays and purified proteins, the authors show that α SNAP competes with the SNARE-organizer Munc18 for binding the target membrane SNARE (t-SNARE) Syntaxin1 binding. In addition α SNAP prevents Syb-2 entry into a t-SNAREs complex of Syntaxin1 and SNAP25, and inhibits fusion of trans-SNARE complexes. These inhibitory functions of α SNAP can be reset by NSF, and fusion can proceed in a Munc13 and Munc18 dependent manner. In addition, the authors show that these inhibitory actions of α SNAP require membrane and SNARE-binding.

These novel inhibitory actions of α SNAP are NSF-independent and therefore distinct from the canonical SNARE-disassembly that α SNAP regulates together with NSF. The reduced model system with synthetic components that the authors have developed over the years is mimicking in situ fusion reactions better and better, e.g., the calcium and Munc18/Munc13 dependence. Their liposome fusion readout systematically contains both lipid- and content-mixing, in accordance with the highest standards in the field. To delineate the different steps and protein conformations, the authors make elegant use of sequential incubation steps and mutant proteins, as well as sensitive readouts such as FRET and NMR. The resulting data is of high quality and convincing. Conceptually, this work builds on their previous Science paper (Ma et al., 2013), in which they introduce the M18/M13 pathway that is resistant to α SNAP-NSF. The proposed model is consistent with the data.

However, it remains unclear to what extent NSF-independent actions of α SNAP are relevant for neuronal secretion and synaptic transmission and to what extent such actions contribute to the inhibition already imposed by the SNARE disassembling activity of NSF- α SNAP.

Major points:

- Physiological relevance. The authors present a detailed description of the different ways α SNAP inhibits SNARE-driven fusion in vitro in an NSF-independent manner. However, the physiological relevance of these actions of α SNAP are not clear. Moreover, for most of the experiments described in the current study, the authors have to remove one of the relevant components of the release machinery to observe the NSF-independent inhibitory effects. In a previous study, a reduction of α SNAP levels by 70% in mouse neurons resulted in only a mild effect on synaptic transmission, and previous in vitro studies using synaptosomes suggested that α SNAP inhibition might not be relevant to neuronal secretion.
- Relevance of α SNAP actions in the absence of NSF. It is not clear whether the inhibitory actions of α SNAP are an NSF-independent function of α SNAP or merely a by-product of NSF-dependent functions: α SNAP must bind to SNAREs in order to recruit of NSF.
- Discrepancy with literature on SNAP-25 binding to Syntaxin-1 containing liposomes. Based on HSQC-NMR in figure S4, the authors conclude that SNAP-25 cannot bind syntaxin-1 when reconstituted in liposomes, most likely because membrane-bound Syntaxin forms tetramers (Fig.8). However, this is in contrast with their previous conclusion that SNAP-25 effectively binds to Syntaxin-1 containing liposomes in co-floitation assays (Ma et al., 2013).
- The experimental design of figure 6. It is not clear why the authors chose to speed up trans-SNARE complex assembly in the assay in figure 6 by including a synaptotagmin-1 fragment and the C-terminal Syb-2 peptide, instead of using M13, especially since they want to test a prediction from the first premise of their model (which is about M13-dependent fusion). The authors should explain their rationale, or repeat the assay with M13 to demonstrate that their conclusions hold under conditions that more closely resemble M13-dependent calcium-independent fusion.

Minor points:

- The black traces in figure 2 and 3 are identical, but are labelled +M13 in figure 2 and +M18 in figure 3.
- In figure 3, the arrows indicating calcium addition are missing.
- Lipid-mixing and content-mixing labels in figure 4 should be on top of panel A and B as in the

previous figures.

- The y-axis of figure 10, panel b would be more informative in different units. In addition, it would be helpful if the main text mentions the color of the trace being discussed, as the labels (V+T) and trans-SC are not very intuitive.
- Error on page 17, line 2: pre-preformed should be pre-formed
- Spelling error on page 21 in the discussion, col-localizing should be co-localizing.

Reviewer #2 (Remarks to the Author):

In this manuscript, Stepien et al describe the effect of alpha-SNAP on reconstituted SNARE-mediated liposome fusion in the presence of various combinations of SNARE assembly and disassembly effector proteins, namely Munc18-1, Munc13-1, and NSF. The major finding of the paper is that aSNAP strongly inhibits fusion by three different mechanisms and that the presence of Munc18-1 can bypass this inhibition, hence rendering the fusion strictly dependent on Munc18-1 and Munc13-1. Within the context of the paper, the presented data appear to support its conclusions. However, the broader significance of the work and its advance relative to previously published work is rather limited for the following reasons.

1. All fusion studies performed in this study monitor calcium-dependent and calcium-independent fusion in the absence of synaptotagmin-1 (and complexin-1). Synaptotagmin-1 has been well established as the calcium sensor of this reaction in vivo and in vitro. The fact that the authors observe calcium independent fusion as well as calcium stimulation of fusion in the absence of synaptotagmin points to the strong possibility that they are focusing in this work on a less relevant side reaction that may not have much in common with the physiological exocytotic fusion reaction. The authors mention on p. 7 that faster assays may be needed to look at the effects of physiological calcium regulation by synaptotagmin (and complexin), which begs the question why this has not been done.

2. The main conclusion that aSNAP inhibits membrane fusion in vitro has been shown and analyzed in clearer molecular mechanistic detail before (Park et al, J. Biol. Chem. 289:16326; 2014), a study that was further preceded by another study on this topic (Barszcewski et al Mol. Biol. Cell 19:776; 2008). The current paper goes beyond these earlier studies by concluding that Munc18-1 has a higher affinity to syntaxin-1 than aSNAP has to syntaxin-1-containing SNARE complexes. This is interesting and could be better stated as the focus of the current work, but it does not rise to the level of a major advance that is broadly significant beyond the specialized field.

Sample statistics: The authors mention that the sample to sample variation between the different pathways is relatively high. This raises questions about the interpretation of the results in the context of the cell. Are these experiments conducted under specialized conditions that give rise to the observed effects or are they broadly robust? The mentioned sample variation should be illustrated and discussed in more detail. The authors should show and discuss the statistics of all of their results and not just those of some examples. If there are indeed high variations from experiment to experiment, three repeats of each condition may not be sufficient. It is also not clear what was repeated: fusion experiments, sample preparations, protein purifications, or reconstitutions?

Reviewer #3 (Remarks to the Author):

In this study, Stepien and colleagues show that aSNAP can inhibit membrane fusion by binding to syntaxin1, t-SNARE complexes and trans SNARE complexes, hindering Munc18-1 binding, precluding further SNARE complex formation and blocking liposome fusion, respectively. This inhibition can be released by NSF in the presence of ATP. In the presence of Munc18-1, membrane fusion becomes aSNAP resistant and the presence of Munc13-1 ensures productive/orderly SNARE

complex assembly.

Although some of the α SNAP interactions have been published previously and a role of Munc18-1 in orderly SNARE complex assembly (including α -SNAP/NSF resistance) has been established, the study clarifies some previous controversies and provides a comprehensive overview how and at which steps α SNAP could inhibit neuronal SNARE complex assembly. Previous studies in vitro and in cellular systems support the physiological relevance of the NSF/ α SNAP controlled SNARE complex assembly pathway. The data are to a large degree convincing, but several technical issues should be resolved.

In general, the authors should show for all experiments how many times (n) the experiments were repeated, including quantifications and corresponding statistical analyses. Presently, this information is available for Figure 2a and b, Figure 5, and Figure 7.

In Figure 6, to which degree does the binding of α SNAP to the t-SNARE interfere with the binding of C2AB to the t-SNARE (SNAP-25) thereby reducing vesicle docking and subsequent trans SNARE complex formation? Can the authors omit the addition of C2AB and Ca^{2+} (e.g. conditions used in Figure 10.)?

The data shown in Figure 9 need quantification and statistical analysis. In particular, binding of α SNAP to syntaxin-1 liposomes is hard to detect. The authors should consider using a different gel system better separating syntaxin-1 from α SNAP (or Western blot analysis employing an anti- α SNAP antibody). Furthermore, in panel a, the addition of 0.75 μM Munc18-1 seems to increase the amount of α SNAP bound to the syntaxin 1 liposomes. Why? In panel b, maximum competition seems to occur already in presence of 0.75 μM α SNAP (3 μM Munc18-1). Increasing concentrations of α SNAP appear to slightly increase the Munc18-1 - syntaxin1 liposome interaction. These issues need to be resolved. Figure 9d is more convincing but the label "NSF" needs to be aligned with the corresponding protein band.

For the comparison of several syntaxin-1 constructs (Figure 5), the authors need to make sure that similar protein and lipid amounts were used in the fusion assay. It should be straight forward to determine the protein lipid ratio for the various SUV preparations. In the case of variations in the reconstitution efficiencies, conditions can be appropriately adjusted to obtain SUVs with comparable syntaxin/lipid ratios.

The authors should also consider to discuss the recent finding that complexin competes with α SNAP for SNARE complex binding (Choi et al., 2018, eLife).

Additional points:

- In the lipid and content mixing assays do the maximum fluorescence signals indeed approach only values below 1 % of max? These signals would be rather minute.
- In the figure legends, please state for clarity the concentrations of the added components (α SNAP, M13, M18) when appropriate.
- In Figure 3, the black fusion curves seem to be mislabeled: M13 should be used instead of M18.

We thank the reviewers for their positive comments and their constructive criticisms. We have tried to address all the concerns with revisions on the manuscript and/or the answers provided below. In the summary of the revisions described below, the comments from the reviewers are in normal font whereas our responses are in blue. Please note also that, as a consequence of the revisions, we have made multiple changes in the numbering of the supplementary figures.

Reviewer #1 (Remarks to the Author):

The manuscript by Stepien et al describes novel mechanisms by which the soluble NSF adaptor protein α SNAP inhibits SNARE-dependent fusion in vitro. Using liposome fusion assays and purified proteins, the authors show that α SNAP competes with the SNARE-organizer Munc18 for binding the target membrane SNARE (t-SNARE) Syntaxin1 binding. In addition α SNAP prevents Syb-2 entry into a t-SNAREs complex of Syntaxin1 and SNAP25, and inhibits fusion of trans-SNARE complexes. These inhibitory functions of α SNAP can be reset by NSF, and fusion can proceed in a Munc13 and Munc18 dependent manner. In addition, the authors show that these inhibitory actions of α SNAP require membrane and SNARE-binding.

These novel inhibitory actions of α SNAP are NSF-independent and therefore distinct from the canonical SNARE-disassembly that α SNAP regulates together with NSF. The reduced model system with synthetic components that the authors have developed over the years is mimicking in situ fusion reactions better and better, e.g., the calcium and Munc18/Munc13 dependence. Their liposome fusion readout systematically contains both lipid- and content-mixing, in accordance with the highest standards in the field. To delineate the different steps and protein conformations, the authors make elegant use of sequential incubation steps and mutant proteins, as well as sensitive readouts such as FRET and NMR. The resulting data is of high quality and convincing. Conceptually, this work builds on their previous Science paper (Ma et al., 2013), in which they introduce the M18/M13 pathway that is resistant to α SNAP-NSF. The proposed model is consistent with the data.

We thank the reviewer for the nice summary of our work.

However, it remains unclear to what extent NSF-independent actions of α SNAP are relevant for neuronal secretion and synaptic transmission and to what extent such actions contribute to the inhibition already imposed by the SNARE disassembling activity of NSF- α SNAP.

Major points:

- **Physiological relevance.** The authors present a detailed description of the different ways α SNAP inhibits SNARE-driven fusion in vitro in an NSF-independent manner. However, the physiological relevance of these actions of α SNAP are not clear. Moreover, for most of the experiments described in the current study, the authors have to remove one of the relevant components of the release machinery to observe the NSF-independent inhibitory effects. In a previous study, a reduction of α SNAP levels by 70%

in mouse neurons resulted in only a mild effect on synaptic transmission, and previous in vitro studies using synaptosomes suggested that α SNAP inhibition might not be relevant to neuronal secretion.

We completely agree that the physiological relevance of our findings remains to be demonstrated. However, many of the fundamental mechanistic advances in our field have emerged from structural and biochemical studies of the interactions between the components of the release machinery performed in vitro (most often with fewer components than those used in the work presented in our paper). By their very nature, all these seminal studies necessarily involved the divide-and-conquer approach where some important components are absent, but it is because of the ability to include or exclude some components that key findings are often made. Demonstrating the physiological relevance of these findings can sometimes be relatively straightforward, but in other cases can take many years. Genetic studies to test the relevance of the NSF-independent functions of α SNAP proposed in our paper provide a clear example of these difficulties because these functions need to be somehow dissected from its function in NSF-dependent disassembly of various types of SNARE complexes, and because there are other SNAP isoforms. We are not aware of any physiological study of α SNAP function that has been sufficiently systematic to allow such dissection.

Importantly, the model that emerges in our paper constitutes a key framework to design these challenging experiments in the future, and at the same time integrates a large amount of available biochemical and physiological data, providing natural explanations for observations that remained enigmatic. Thus, SNARE complexes can assemble without the need of Munc18-1 and Munc13s, large amounts of syntaxin-1-SNAP-25 complexes likely exist on the plasma membrane given the abundance of these proteins, and t-SNAREs from plasma membrane lawns readily form complexes with exogenous synaptobrevin (ref. 35). And yet, release is completely abrogated in the absence of Munc18-1 and Munc13s (refs. 23, 24). How is the system guided to the Munc18-1-Munc13 pathway, preventing non-regulated fusion independent of Munc18-1 and Munc13s? These findings might be explained by the observation that trans-SNARE complexes are disassembled by NSF- α SNAP, and that Munc18-1+Munc13-1 provide an NSF- α SNAP resistant pathway to trans-SNARE complex assembly (refs. 36,38). However, the Jahn lab and ours observed that some trans-SNARE complexes are not disassembled by NSF- α SNAP (ref.s 36, 38). Moreover, while the functions of Munc18-1 and Munc13s are clearly related, multiple observations indicated that there is an activity of Munc18-1 that renders this protein more essential than Munc13s, including the fact that multiple means have been found to overcome the absence of Munc13s but not of Munc18-1, and the finding by the Verhage lab that inactivation of NSF by N-ethylmaleimide can partially enable neurotransmitter release in Munc13-1/2 DKO neurons but not in the Munc18-1 KO (ref. 37).

All of these findings can now be rationalized by our model postulating that α SNAP prevents SNARE-dependent membrane fusion by multiple mechanisms and that this inhibition can only be overcome when Munc18-1 binds to closed syntaxin-1. We would like to emphasize that an important feature of this model is that it makes a lot of sense based on the functional, structural and biochemical properties of the proteins involved. Moreover, over the years we have established multiple correlations between the effects of mutations on our reconstitution assays and the effects of the same

mutations on neurotransmitter release in neurons (refs. 16, 18, 19, 29, 51). Hence, our reconstitutions provide a powerful system to understand the molecular events that lead to neurotransmitter release and to postulate new models that can later be tested by physiological studies.

- Relevance of α SNAP actions in the absence of NSF. It is not clear whether the inhibitory actions of α SNAP are an NSF-independent function of α SNAP or merely a by-product of NSF-dependent functions: α SNAP must bind to SNAREs in order to recruit NSF.

We agree that this issue is not clear. However, in order to understand how this complex system is controlled, it is important to know that α SNAP can inhibit fusion by itself, in fact by three NSF-independent mechanisms. Since α SNAP must bind to SNARE four-helix bundles first, before the 20S complex including NSF can be assembled, it is natural to expect that α SNAP binding to the four-helix bundle (either syntaxin-1-SNAP-25 heterodimers or trans-SNARE complexes) provides a first 'stop-gap' measure to prevent fusion before NSF can be recruited for disassembly (as mentioned in the discussion, middle of page 21). It is also important to note that the fact that the inhibitory functions of α SNAP are NSF-independent and the finding that the inhibitory activities can only be overcome if Munc18-1 binds to closed syntaxin-1 provide a sensible model to rationalize the observation that inactivation of NSF with NEM enables neurotransmitter release in Munc13-1/2 DKO mice but not in Munc18-1 KO mice (ref. 37).

- Discrepancy with literature on SNAP-25 binding to Syntaxin-1 containing liposomes. Based on HSQC-NMR in figure S4, the authors conclude that SNAP-25 cannot bind syntaxin-1 when reconstituted in liposomes, most likely because membrane-bound Syntaxin forms tetramers (Fig.8). However, this is in contrast with their previous conclusion that SNAP-25 effectively binds to Syntaxin-1 containing liposomes in collocation assays (Ma et al., 2013).

Thank you for pointing out what indeed appears to be a contradiction. The different results likely arise because we are now purifying syntaxin-1 in dodecylphosphocholine, which prevents formation of large syntaxin-1 aggregates (ref. 59). We now point out this possibility in the bottom of page 12. Note that this paper also found that syntaxin-1 lacking the N-terminal region, purified by this method and incorporated into micelles, also did not bind to SNAP-25.

- The experimental design of figure 6. It is not clear why the authors chose to speed up trans-SNARE complex assembly in the assay in figure 6 by including a synaptotagmin-1 fragment and the C-terminal Syb-2 peptide, instead of using M13, especially since they want to test a prediction from the first premise of their model (which is about M13-dependent fusion). The authors should explain their rationale, or repeat the assay with M13 to demonstrate that their conclusions hold under conditions that more closely resemble M13-dependent calcium-independent fusion.

We agree that it is better to perform these assays with the Munc13-1 C₁C₂BMUNC₂C fragment instead of synaptotagmin-1 C₂AB. We have performed the experiments with

Munc13-1 C₁C₂BMUNC₂C and obtained analogous results, which are now presented in the revised Fig. 6.

Minor points:

- The black traces in figure 2 and 3 are identical, but are labelled +M13 in figure 2 and +M18 in figure 3.

We have corrected the mistake in Fig. 3, where the black trace is now labeled +M13.

- In figure 3, the arrows indicating calcium addition are missing.

We have included the arrows in the revised Fig. 3.

- Lipid-mixing and content-mixing labels in figure 4 should be on top of panel A and B as in the previous figures.

We have included the lipid mixing and content mixing labels in the revised Fig. 4.

- The y-axis of figure 10, panel b would be more informative in different units. In addition, it would be helpful if the main text mentions the color of the trace being discussed, as the labels (V+T) and trans-SC are not very intuitive.

We are not sure what kind of units the reviewer would like us to use; it is not uncommon to present fluorescence spectra in arbitrary units. In the revised manuscript we now mention the color of the traces being discussed.

- Error on page 17, line 2: pre-preformed should be pre-formed

We have corrected the mistake.

- Spelling error on page 21 in the discussion, col-localizing should be co-localizing.

We have corrected the mistake.

Reviewer #2 (Remarks to the Author):

In this manuscript, Stepien et al describe the effect of alpha-SNAP on reconstituted SNARE-mediated liposome fusion in the presence of various combinations of SNARE assembly and disassembly effector proteins, namely Munc18-1, Munc13-1, and NSF. The major finding of the paper is that aSNAP strongly inhibits fusion by three different mechanisms and that the presence of Munc18-1 can bypass this inhibition, hence rendering the fusion strictly dependent on Munc18-1 and Munc13-1. Within the context of the paper, the presented data appear to support its conclusions. However, the broader significance of the work and its advance relative to previously published work is rather limited for the following reasons.

1. All fusion studies performed in this study monitor calcium-dependent and calcium-independent fusion in the absence of synaptotagmin-1 (and complexin-1). Synaptotagmin-1 has been well established as the calcium sensor of this reaction in vivo and in vitro. The fact that the authors observe calcium independent fusion as well as calcium stimulation of fusion in the absence of synaptotagmin points to the strong possibility that they are focusing in this work on a less relevant side reaction that may not have much in common with the physiological exocytotic fusion reaction. The authors mention on p. 7 that faster assays may be needed to look at the effects of physiological calcium regulation by synaptotagmin (and complexin), which begs the question why this has not been done.

We respectfully disagree with the logic of this criticism. Establishing correlations with physiological data is key to assess the likelihood that reconstitution experiments recapitulate to some degree the molecular events that occur in vivo. Electrophysiological studies with knockout mice revealed that all forms of neurotransmitter release (evoked, spontaneous and sucrose-induced release) were abolished in the absence of Munc18-1 or Munc13-1/2 (refs. 23,24), whereas milder phenotypes were observed in synaptotagmin-1 KO mice even upon knockdown of synaptotagmin-7, as evoked release was strongly impaired but substantial spontaneous and sucrose-induced release remained [Bacaj et al. *Neuron* **80**, 947 (2013); Bacaj et al. *PLoS. Biol* **13**, e1002267 (2015)]. Similar phenotypes were observed in complexin-1/2/3 TKO mice, although evoked release was less impaired in this case [Xue et al. *Proc. Natl. Acad. Sci. U. S. A* **105**, 7875 (2008); Xue et al. *Nat. Struct. Mol. Biol* **17**, 568 (2010)]. Hence, the physiological data suggest that Munc18-1 and Munc13s are more critical for synaptic vesicle fusion per se than synaptotagmin-1 and complexins, although there is little doubt that synaptotagmin-1 is crucial to accelerate fusion and thus allow release in less than 1 ms after Ca^{2+} influx into a presynaptic terminal (in the ms time scale).

Based on these findings, the observation of membrane fusion in our assays without synaptotagmin-1 (or complexin-1) is perfectly compatible with the available physiological data. We are not sure whether the comments from the reviewer arise in part from the multiple reconstitution studies showing that synaptotagmin-1 enhances SNARE-dependent liposome fusion, but it is important to note that these studies did have an important disagreement with physiological data, as they observed fusion without Munc18-1 and Munc13s. This does not necessarily imply that the fusion observed in those studies reflected a 'less relevant side reaction', but it certainly does not seem logical to conclude that our fusion assays, which are more complete and incorporate the most crucial components of the release machinery, are the ones that focus on a less relevant side reaction. It is also important to note that our reconstitutions not only recapitulate the crucial requirement of Munc18-1 and Munc13-1 for fusion, but in addition correlate in many aspects with physiological data obtained through extensive mutagenesis studies (refs. 16, 18, 19, 29, 51).

With regard to why we have not yet addressed the question as to whether synaptotagmin-1 accelerates fusion at faster time scales, this is not as trivial issue. Our fusion assays exhibit a strong Ca^{2+} dependence that arises because of Ca^{2+} binding to the Munc13-1 C₂B domain. Physiological data have shown that Ca^{2+} binding to this domain plays an important function in neurotransmitter release; some evidence suggest a role in

the Ca²⁺-triggering step of release while other results point to a facilitation of priming during repetitive stimulation [Shin et al. *Nat. Struct. Mol. Biol* **17**, 280 (2010); Michelassi et al. *Neuron* **95**, 577 e5 (2017)], and it is perfectly plausible that Ca²⁺ binding to the Munc13-1 C₂B domain in fact has a role in both. Hence, the Ca²⁺ dependence of liposome fusion that we observe may reflect an overlap of functions in priming and in fusion that cannot be resolved in the second time scale of our fusion assays. Because fusion occurs so quickly in this time scale upon Ca²⁺ addition in our standard conditions (e.g. Fig. 4b), it is impossible to observe a Ca²⁺-dependent stimulatory effect of synaptotagmin-1 even if it exists.

[redacted]

We realize that the sentence where we explained that synaptotagmin-1 and complexin-1 had no effect in our fusion assays may have been somewhat confusing. We have rephrased this statement, which now reads (end of page 7):

‘We also tested whether inclusion of synaptotagmin-1 into the synaptobrevin-containing liposomes or addition of complexin-1 altered the results of these assays, but we did not observe any noticeable effects (Supplementary Fig. 2). Note that it is plausible that synaptotagmin-1 and/or complexin-1 might accelerate the rate of fusion but such acceleration cannot be detected on the second time scale characteristic of these bulk assays. Regardless of this possibility, these results show that synaptotagmin-1 and complexin-1 cannot overcome the inhibition of fusion caused by α SNAP.’

2. The main conclusion that α SNAP inhibits membrane fusion in vitro has been shown and analyzed in clearer molecular mechanistic detail before (Park et al, J. Biol. Chem. 289:16326; 2014), a study that was further preceded by another study on this topic (Barszcewski et al Mol. Biol. Cell 19:776; 2008). The current paper goes beyond these earlier studies by concluding that Munc18-1 has a higher affinity to syntaxin-1 than α SNAP has to syntaxin-1-containing SNARE complexes. This is interesting and could be better stated as the focus of the current work, but it does not rise to the level of a major advance that is broadly significant beyond the specialized field.

We agree that these two papers from the Jahn lab analyzed some aspects of the inhibition by α SNAP in more detail than we did; in fact, it was our intention not to analyze details that were already published in those papers. However, we cannot agree with the statement that the mechanistic details were clearer in those papers, as the picture that emerged from them was somewhat confusing because different mechanisms of inhibition were proposed in the two papers and the data were not integrated into a coherent model like the one we present in our paper. Moreover, those two papers did not analyze the interplay of α SNAP with Munc18-1 and Munc13-1, and most of the data using reconstituted proteoliposomes was obtained with a syntaxin-1 fragment lacking the N-terminal region, whereas all of our data was obtained with full-length syntaxin-1 except for the experiments where we tested the effects of deleting the H_{abc} domain.

We would like to point out that we sent the paper to Reinhard Jahn and William Wickner before submitting it for publication because we wanted to know their opinion on our study, given their important contributions to understand the roles of α SNAP and its yeast homologue Sec17. Both colleagues had high praise for our study and, among several other laudatory comments, Reinhard wrote that 'the results were very interesting and the findings very clear'. We are sorry that we neglected to acknowledge their advice in the original manuscript and have corrected this mistake in the revised paper, where we include a note thanking both colleagues in the acknowledgments.

In any case, we do agree with the reviewer that the results with Munc18-1 are the more novel aspect of the paper and have changed the title of the revised manuscript to shift the focus on this aspect, although in the abstract and the text we still describe our results on the inhibitory activity of α SNAP by three distinct mechanisms because they are critical to develop the integrated model that we present, as well as to underline the importance of Munc18-1 binding to closed syntaxin-1 to overcome the α SNAP inhibition.

Sample statistics: The authors mention that the sample to sample variation between the different pathways is relatively high. This raises questions about the interpretation of the results in the context of the cell. Are these experiments conducted under specialized conditions that give rise to the observed effects or are they broadly robust? The mentioned sample variation should be illustrated and discussed in more detail. The authors should show and discuss the statistics of all of their results and not just those of some examples. If there are indeed high variations from experiment to experiment, three repeats of each condition may not be sufficient. It is also not clear what was

repeated: fusion experiments, sample preparations, protein purifications, or reconstitutions?

In general, there were no high variations from experiment to experiment as the reviewer seems to imply. There was substantial variability in the results obtained with different preparations for the specific issue that we mentioned, i.e. the relative amplitude of the Ca^{2+} -independent and Ca^{2+} -dependent components of membrane fusion observed between T- and V-liposomes in the presence of Munc18-1 and Munc13-1 C₁C₂BMUNC₂C (green curves in Fig. 2a,b). From the overall study it became clear that Ca^{2+} -independent fusion in these experiments is mediated by syntaxin-1-SNAP-25 heterodimers whereas the Ca^{2+} -dependent component arises from a population of isolated syntaxin-1 in the T-liposomes that can bind to Munc18-1 and that is variable in different preparations. Such variability most likely arises because of partial aggregation, and hence is difficult to control.

To illustrate this variability, in the revised manuscript we have included results from an additional preparation in Supplementary Fig. 1c,d, and we cite another data set under the same conditions that was published previously (dark blue traces in Fig. 5C,D of ref. 18). We also include a Supplementary Table in Excel format summarizing the statistics for pairwise comparisons among the results obtained for each quantitative data set shown in the figures, and in the methods section we clarify that 'All experiments were repeated at least three times with a given preparation and the results were verified in multiple experiments performed with different preparations.' (bottom of page 24).

Reviewer #3 (Remarks to the Author):

In this study, Stepien and colleagues show that α SNAP can inhibit membrane fusion by binding to syntaxin1, t-SNARE complexes and trans SNARE complexes, hindering Munc18-1 binding, precluding further SNARE complex formation and blocking liposome fusion, respectively. This inhibition can be released by NSF in the presence of ATP. In the presence of Munc18-1, membrane fusion becomes α SNAP resistant and the presence of Munc13-1 ensures productive/orderly SNARE complex assembly.

Although some of the α SNAP interactions have been published previously and a role of Munc18-1 in orderly SNARE complex assembly (including α -SNAP/NSF resistance) has been established, the study clarifies some previous controversies and provides a comprehensive overview how and at which steps α SNAP could inhibit neuronal SNARE complex assembly. Previous studies in vitro and in cellular systems support the physiological relevance of the NSF/ α SNAP controlled SNARE complex assembly pathway. The data are to a large degree convincing, but several technical issues should be resolved.

We thank the reviewer for the nice summary of our results.

In general, the authors should show for all experiments how many times (n) the experiments were repeated, including quantifications and corresponding statistical

analyses. Presently, this information is available for Figure 2a and b, Figure 5, and Figure 7.

As mentioned above in the response to the comment on statistics made by reviewer 2, in the revised manuscript we provide additional details on how we verified the reproducibility of our results and we include a Supplementary Table in Excel format summarizing the statistics for pairwise comparisons among the results obtained for each quantitative data set shown in the figures. We have quantified the results of Fig. 3 and, hence, quantitative analyses are now shown for most the reconstitution experiments that compared the lipid and mixing observed with different sets of reagents (Figs. 2a,b, 3, 5 and 7). However, we believe that authors are often placing too much weight on analysis of statistical significance without really stopping to think about the actual meaning of the statistical analysis and the need perform it, an opinion that is shared by many [e.g. Amrhein et al. *Nature* **567**, 305 (2019)]. For instance, we believe that the results of Figs. 4a,b and 9c,d,f are clearcut and no statistical analysis is necessary in this case, although we will provide such analysis if it is absolutely required for acceptance of the paper for publication.

In Figure 6, to which degree does the binding of α SNAP to the t-SNARE interfere with the binding of C2AB to the t-SNARE (SNAP-25) thereby reducing vesicle docking and subsequent trans SNARE complex formation? Can the authors omit the addition of C2AB and Ca^{2+} (e.g. conditions used in Figure 10.)?

As mentioned in our response to one of the minor concerns of reviewer #1, we agree that using synaptotagmin-1 C₂AB for these experiments was a poor choice and we have now performed analogous experiments where instead we used Munc13-1 C₁C₂BMUNC₂C to facilitate trans-SNARE complex assembly (revised Fig. 6).

The data shown in Figure 9 need quantification and statistical analysis. In particular, binding of α SNAP to syntaxin-1 liposomes is hard to detect. The authors should consider using a different gel system better separating syntaxin-1 from α SNAP (or Western blot analysis employing an anti- α SNAP antibody). Furthermore, in panel a, the addition of 0.75 μ M Munc18-1 seems to increase the amount of α SNAP bound to the syntaxin 1 liposomes. Why? In panel b, maximum competition seems to occur already in presence of 0.75 μ M α SNAP (3 μ M Munc18-1). Increasing concentrations of α SNAP appear to slightly increase the Munc18-1 - syntaxin1 liposome interaction. These issues need to be resolved. Figure 9d is more convincing but the label “NSF” needs to be aligned with the corresponding protein band.

Liposome co-floatation assays are not readily amenable to quantification because it is difficult to consistently collect the entire layer containing the liposomes and only that layer. This is the reason why the results observed in some of the lanes did not fit perfectly with the rest of the data in the specific lanes mentioned by the reviewer. Moreover, there are unavoidable differences in the degree of staining among different gels and the band intensity does not increase completely linearly with the amount of protein, which hinders quantitative comparisons of data from different gels. However, firm conclusions can be drawn from these experiments by looking at the overall data

and making sure that the results are reproducible. Nevertheless, we agree with the reviewer that these experiments would be more conclusive if we could separate the bands from α SNAP and syntaxin-1. We have repeated multiple times the same experiments but using α SNAP that contained a flexible C-terminal sequence from an uncleaved His₆-tag and can be readily separated from syntaxin-1 by SDS-PAGE. The results are presented in the revised Fig. 9 and fully confirm the conclusions, namely that α SNAP and Munc18-1 compete for binding to syntaxin-1, and that Munc18-1 displaces α SNAP that is pre-bound to syntaxin-1 if samples are incubated for a sufficiently long time or NSF is included in the reaction. We hope that the reviewer will agree that the results are sufficiently clearcut and no quantification is required.

For the comparison of several syntaxin-1 constructs (Figure 5), the authors need to make sure that similar protein and lipid amounts were used in the fusion assay. It should be straight forward to determine the protein lipid ratio for the various SUV preparations. In the case of variations in the reconstitution efficiencies, conditions can be appropriately adjusted to obtain SUVs with comparable syntaxin/lipid ratios.

We thank the reviewer for raising this important point and indeed there were some differences in the amounts of the different syntaxin-1 fragments that were reconstituted in the liposomes. However, the protein amounts were comparable for full-length syntaxin-1 and the syntaxin-1 Δ H_{abc} fragment (revised Supplementary Fig. 4c). Since the results obtained with this fragment were the most critical for the conclusions we drew from these data and we wanted to shorten a little the long manuscript, in the revised version we have removed the data obtained with the Δ N_{pep} and LE mutant syntaxin-1, and present only the data comparing the results obtained with full-length and Δ H_{abc} syntaxin-1 (revised Fig. 5).

The authors should also consider to discuss the recent finding that complexin competes with α SNAP for SNARE complex binding (Choi et al., 2018, eLife).

We now cite this finding and the corresponding reference in the revised manuscript (bottom of page 20).

Additional points:

- In the lipid and content mixing assays do the maximum fluorescence signals indeed approach only values below 1 % of max? These signals would be rather minute.

Thank you for pointing this out. We wrote %max on the y-axis but the data were actually normalized to 1 rather than 100. We have corrected all the y-axes to the actual %max values for all the corresponding figures.

- In the figure legends, please state for clarity the concentrations of the added components (α SNAP, M13, M18) when appropriate.

We have added this information in the figure legends.

- In Figure 3, the black fusion curves seem to be mislabeled: M13 should be used instead of M18.

We have corrected this mistake.

We hope that we have properly addressed all the concerns and that the paper is now acceptable for publication in its current form. We would like to thank the reviewers again for their valuable comments.

Sincerely,

Josep Rizo

Reviewers' comments:

Reviewer #1 (Remarks to the Author):

The authors addressed two of the main issues raised, one textual (#3) and one with a new liposome fusion experiment (#4). With this, these issues are covered adequately.

My main criticism of this study (major points #1 and #2) remains.

The physiological relevance of the main findings in this study remains uncertain (major point #1) and previously published physiological experiments lowering α SNAP levels report no effects consistent with the inhibitory role proposed by the authors. The authors argue that "fundamental mechanistic advances have emerged from structural and biochemical studies [] performed in vitro" and that their study is a step beyond where the field is today. Both arguments are correct. Subsequently, the authors argue that physiological studies of α SNAP function have not been sufficiently systematic to argue against their proposed mechanism, also given the fact that α SNAP/NSF have multiple roles (in the model of the authors). This is also true, but this reviewer believes it is the responsibility of the authors to prove relevance for a paper in a top journal.

The relevance of α SNAP actions in the absence of NSF (major point #2) also remains unresolved. The authors basically agree ("We agree that this issue is not clear"). The authors state that they "provide a sensible model". This is true. However, in practice the two molecules are present together in the synapse and it remains unclear what the relevance is of experiments where NSF is omitted.

Reviewer #3 (Remarks to the Author):

In their revised manuscript, Stepien and colleagues have to a large degree satisfactorily addressed my previous concerns.

However, although in the revised Figure 9c, the data clearly show a competition between α SNAP and Munc18-1, the stoichiometries/ratios to syntaxin-1 seem to be off-scale. This is also the case for Fig. 9d, 15 μ M Munc18 (protein aggregates?). Considering that only 50% of syntaxin-1 (facing the outside of the SUVs) should be accessible, there seems to be a significant excess of α SNAP and Munc18-1 bound to the S liposomes. Thus, it is unclear if the authors indeed measure competitive binding to syntaxin-1. Figures 9a and 9b seem to be okay. Please clarify.

We thank the reviewers for their additional comments. We have tried to address the concerns with revisions on the manuscript and/or the answers provided below. In the summary of the revisions described below, the comments from the reviewers are in normal font whereas our responses are in blue.

Reviewer #1 (Remarks to the Author):

The authors addressed two of the main issues raised, one textual (#3) and one with a new liposome fusion experiment (#4). With this, these issues are covered adequately.

My main criticism of this study (major points #1 and #2) remains.

The physiological relevance of the main findings in this study remains uncertain (major point #1) and previously published physiological experiments lowering aSNAP levels report no effects consistent with the inhibitory role proposed by the authors. The authors argue that “fundamental mechanistic advances have emerged from structural and biochemical studies [] performed in vitro” and that their study is a step beyond where the field is today. Both arguments are correct. Subsequently, the authors argue that physiological studies of aSNAP function have not been sufficiently systematic to argue against their proposed mechanism, also given the fact that aSNAP/NSF have multiple roles (in the model of the authors). This is also true, but this reviewer believes it is the responsibility of the authors to prove relevance for a paper in a top journal.

We are glad that the reviewer agrees with our scientific arguments and we respect the opinion that the authors should prove relevance for a paper in a top journal. However, this standard has not always been applied by top journals such as Nature, various other Nature journals, Science, Cell and eLife, among many others, which have published many papers describing important findings in our field without demonstrating physiological relevance. Multiple examples of this fact can be found in the reference list of our paper and include seminal contributions that relied mostly on reconstitution approaches and did not present any physiological data.

The relevance of aSNAP actions in the absence of NSF (major point #2) also remains unresolved. The authors basically agree (“We agree that this issue is not clear”). The authors state that they “provide a sensible model”. This is true. However, in practice the two molecules are present together in the synapse and it remains unclear what the relevance is of experiments where NSF is omitted.

The issue again revolves around physiological relevance. As described in the introduction of our article, previously published physiological data suggested that there is an NSF-independent pathway that completely prevents synaptic vesicle fusion in the absence of Munc18-1. The in vitro results described in our paper now provide a very plausible molecular mechanism to explain these physiological findings. Their relevance is in addition supported by how this mechanism fits with a large amount of biochemical and structural data available on these proteins, and by the dramatic effects that we

observed, both in terms of the inhibitory function of α SNAP on fusion and how Munc18-1 overcomes this inhibition. Publication of our results will allow this model to be further tested by other laboratories that are much better prepared than us to perform experiments in neurons.

Reviewer #3 (Remarks to the Author):

In their revised manuscript, Stepien and colleagues have to a large degree satisfactorily addressed my previous concerns. However, although in the revised Figure 9c, the data clearly show a competition between alpha SNAP and Munc18-1, the stoichiometries/ratios to syntaxin-1 seem to be off-scale. This is also the case for Fig. 9d, 15 microM Munc18 (protein aggregates?). Considering that only 50% of syntaxin-1 (facing the outside of the SUVs) should be accessible, there seems to be a significant excess of alphaSNAP and Munc18-1 bound to the S liposomes. Thus, it is unclear if the authors indeed measure competitive binding to syntaxin-1. Figures 9a and 9b seem to be okay. Please clarify.

We thank the reviewer for pointing out these issues, which indeed need an explanation. As we point out in response to other concerns about these experiments from the first round of reviews, the conclusions rely on the trends observed in the overall data rather than on quantitative details of individual experiments. Obtaining quantitative data using co-floatation assays is hindered by technical limitations of the technique (such as the difficulty of pipetting the entire layer containing the liposomes and only that layer, as well as by the lack of linearity in the band intensities of Coomassie blue stained gels). Some variability in the results may also arise from the tendency of syntaxin-1 to aggregate, as exemplified by single molecule studies from the laboratory of Axel Brunger where syntaxin-1-to-lipid ratios of 1:10,000,000 were used to have at least 50% of monomeric syntaxin-1 in lipid bilayers [Weninger et al. *Structure* **16**, 308-320 (2008)]. Although we used a new method to try to minimize syntaxin-1 aggregation during purification, we cannot rule out that some of the syntaxin-1 may be aggregated within the reconstituted proteoliposomes and that such aggregation may occur to different extents in different liposome preparations. We also note that in our hands only about 30-40% of syntaxin-1 is located in the lumen of the liposomes and that Munc18-1 is larger than syntaxin-1; hence, the stronger intensity of the Munc18-1 band is not unexpected. However, we do agree that it is surprising that the α SNAP band was stronger than that of syntaxin-1 in Fig. 9c. We have observed weaker intensity of syntaxin-1 from proteoliposome samples compared to samples in detergent, but we have not done a sufficiently systematic study to draw definitive conclusions about this issue.

To address the concern from the reviewer, we have replaced the gel of Fig. 9c with a gel from an analogous experiment. The gel from the previous Fig. 9c is now shown in panel c of a new Supplementary Figure (Supplementary Fig. 8), which also displays additional experiments performed under conditions analogous to those of Fig. 9a,b,d. This figure

illustrates the variability that we observed in the co-floatation assays and at the same time the consistency in the overall data, which supports our conclusions. In the legend of this supplementary figure we point out both the variability and the fact that the conclusions are supported by the overall trends despite the variability. We have also added the following sentence in the main text (bottom of page 15):

‘We note that, because of technical difficulties, there was a natural variability among the results obtained in different experiments performed under the same conditions, but the trends observed in the overall data yielded clear conclusions (Fig. 9, Supplementary Fig. 8a-d).’

In the legend of Figure 9, we have added a sentence regarding the strong Munc18-1 band in the 15 mM sample of panel d:

‘Note that the strong band of Munc18-1 in the lane corresponding to 15 mM Munc18-1 may arise from aggregation and/or weak binding to the liposomes. Additional co-floatation assays performed under analogous conditions are shown in Supplementary Fig. 8a-d, where other technical details that need to be considered about these assays are discussed.’

In addition, we have performed a gel filtration assay that confirms the competition between Munc18-1 and α SNAP for binding to liposome-anchored syntaxin-1, as well as the conclusion that Munc18-1 displaces α SNAP from syntaxin-1 if incubated for a sufficiently long time or if added first (Supplementary Fig. 8e,f). These conclusions stand on firm grounds, as they are now supported by results obtained with gel filtration, co-floatation assays, liposome fusion assays (Fig. 7) and NMR spectroscopy (Fig. 8, Supplementary Fig. 7).

REVIEWERS' COMMENTS:

Reviewer #3 (Remarks to the Author):

In their second revision, Stepien and colleagues show additional co-floatation data and now include gel filtration experiments to demonstrate the competitive binding between alpha-SNAP and Munc18-1 to syntaxin-1 liposomes. The gel filtration data support the competition.

The binding studies using liposomes are still the weakest part of the manuscripts and are less convincing than the rest of the paper. The authors mention this problem in their manuscript ("trends").

In particular, alpha-SNAP binding to the syntaxin-1 liposomes seems to be highly variable. In the previous Fig. 9c (now suppl. Fig. 8, showing a clear competition), an excess of alpha-SNAP was bound to syntaxin-1 indicating the presence of low affinity "unspecific" liposome interactions (may be via the alpha-SNAP hydrophobic loop) or the presence of alpha-SNAP aggregates binding to syntaxin-1, causing background problems. Comparing the 0 h and the 20 h time points in the current Figure 9c, despite the obvious Munc18-1 binding a reduction of alpha-SNAP binding is barely detectable. In this experiment alpha SNAP binding to syntaxin-1 is substoichiometric, which may also explain the lack of a prominent competition. Part of this problem may be caused by syntaxin-1 existing in different states in the liposome preparations ("close" monomers or "open" oligomers), affecting alpha SNAP binding. The addition of NSF seems to resolve some of these issues (Figure 9d), and the competition becomes more obvious, but unexpectedly NSF co-floats with liposomes. In the long run, the authors need to increase the robustness of their assay, which may be technically challenging. The actual fractionation of the gradients is technically feasible and recovering reproducibly similar liposome amounts should not be a problem.

We thank the reviewer for the additional comments. Our response below is in blue.

Reviewer #3 (Remarks to the Author):

In their second revision, Stepien and colleagues show additional co-floatation data and now include gel filtration experiments to demonstrate the competitive binding between alpha-SNAP and Munc18-1 to syntaxin-1 liposomes. The gel filtration data support the competition.

The binding studies using liposomes are still the weakest part of the manuscripts and are less convincing than the rest of the paper. The authors mention this problem in their manuscript (“trends”).

In particular, alpha-SNAP binding to the syntaxin-1 liposomes seems to be highly variable. In the previous Fig. 9c (now suppl. Fig. 8, showing a clear competition), an excess of alpha-SNAP was bound to syntaxin-1 indicating the presence of low affinity “unspecific” liposome interactions (may be via the alpha-SNAP hydrophobic loop) or the presence of alpha-SNAP aggregates binding to syntaxin-1, causing background problems. Comparing the 0 h and the 20 h time points in the current Figure 9c, despite the obvious Munc18-1 binding a reduction of alpha-SNAP binding is barely detectable. In this experiment alpha SNAP binding to syntaxin-1 is substoichiometric, which may also explain the lack of a prominent competition. Part of this problem may be caused by syntaxin-1 existing in different states in the liposome preparations (“close” monomers or “open” oligomers), affecting alpha SNAP binding. The addition of NSF seems to resolve some of these issues (Figure 9d), and the competition becomes more obvious, but unexpectedly NSF co-floats with liposomes. In the long run, the authors need to increase the robustness of their assay, which may be technically challenging. The actual fractionation of the gradients is technically feasible and recovering reproducibly similar liposome amounts should not be a problem.

We agree that the co-floatation assays are not the strongest part of the paper and in the revised manuscript we acknowledged the pitfalls of these assays, providing two data sets for each group of experiments to illustrate the variability in the results and describing potential reasons for this variability. These reasons can also underlie most of the issues raised by the reviewer in this third round of review. There is a new comment on the observation of the NSF band in panel d, which was present in all the gels that we presented in the different versions of the manuscript that we have submitted, including the original. The band likely arises because of some weak affinity of NSF for the proteoliposomes, and its observation does not affect the conclusions of the experiments. We still believe that the results of these co-floatation assays support the overall conclusions that we draw about the competition between Munc18-1 and α SNAP for syntaxin-1 binding, which are further supported by the reconstitution, NMR and gel filtration experiments. We present an honest account of the data and the pitfalls of the experiments, so that readers can judge by themselves.

There was no specific recommendation by the reviewer on how to revise the manuscript. In the spirit of the overall concern about these assays, we have toned down a sentence where we stated 'but the trends observed in the overall data yielded clear conclusions', which now reads 'but the trends observed in the overall data supported the conclusions from the reconstitution and NMR experiments ...' (middle of page 15). In addition, we have moved the gel filtration data from Supplementary Fig. 8 to Fig. 9 to emphasize these data, which support the competition as pointed out by the reviewer.